# Interpreting Arithmetic Reasoning in Large Language Models using Game-Theoretic Interactions

**Leilei Wen[1]**   **Liwei Zheng[1]**   **Hongda Li[1]**   **Lijun Sun[1]**   **Zhihua Wei[1]**   **Wen Shen[1]***

[1]Tongji University, Shanghai, China

{2331934, 2432010, 2410907, sunlijun, zhihua_wei, wenshen}@tongji.edu.cn

## Abstract

In recent years, large language models (LLMs) have made significant advancements in arithmetic reasoning. However, the internal mechanism of how LLMs solve arithmetic problems remains unclear. In this paper, we propose explaining arithmetic reasoning in LLMs using game-theoretic interactions. Specifically, we disentangle the output score of the LLM into numerous interactions between the input words. We quantify different types of interactions encoded by LLMs during forward propagation to explore the internal mechanism of LLMs for solving arithmetic problems. We find that (1) the internal mechanism of LLMs for solving simple one-operator arithmetic problems is their capability to encode operand-operator interactions and high-order interactions from input samples. Additionally, we find that LLMs with weak one-operator arithmetic capabilities focus more on background interactions. (2) The internal mechanism of LLMs for solving relatively complex two-operator arithmetic problems is their capability to encode operator interactions and operand interactions from input samples. (3) We explain the task-specific nature of the LoRA method from the perspective of interactions.

## 1   Introduction

In recent years, the arithmetic reasoning capabilities of large language models (LLMs) have improved significantly, but the internal mechanism is still unclear. Some studies identified neurons that have great effects on arithmetic reasoning [Yu and Ananiadou, 2024, Rai and Yao, 2024]. Other studies evaluated the impact of modifying words in input arithmetic questions on neuron activations and network outputs [Stolfo et al., 2023, Zhang et al., 2024]. However, previous studies have not mathematically guaranteed that the explanations faithfully reflect the arithmetic reasoning logic of LLMs. Recently, a series of studies have used game-theoretic interactions between input variables to explain the representation power of traditional DNNs [Ren et al., 2024, Deng et al., 2024], e.g., the interaction between "*green*" and "*hand*" forms the concept of "*beginner*." The interaction has been proven to be faithful explanations by a series of theoretical guarantees [Ren et al., 2023a].

Inspired by these studies, we use interactions to explain the arithmetic reasoning capability of LLMs. As Figure 1 shows, given an input arithmetic question $\boldsymbol{x}$, e.g., "*What is 2 plus 7? Answer is,*" the interaction $S$ (e.g., $S = \{2, plus, 7\}$) represents an AND relationship between the words in $S$, which is **equivalently**[2] encoded by the LLM. Each interaction $S$ contributes a numerical effect $I_S$ to push the output score towards generating the answer "*9.*" Thus, we can construct a logic model $\phi(\boldsymbol{x})$ based on interactions. The faithfulness of using the logical model to explain an LLM is reflect as follows.

• As Figure 1(a) shows, given the same input sentence $\boldsymbol{x}$, the output score of the LLM is equal to the output of the logical model, that is, $v(\boldsymbol{x}) = \phi(\boldsymbol{x}) = \sum_{S \subseteq N} I_S$, where $N$ is the set containing all the

---

*Corresponding author.

[2]Note that each interaction is equivalently encoded by the entire DNN, rather than by a specific neuron.

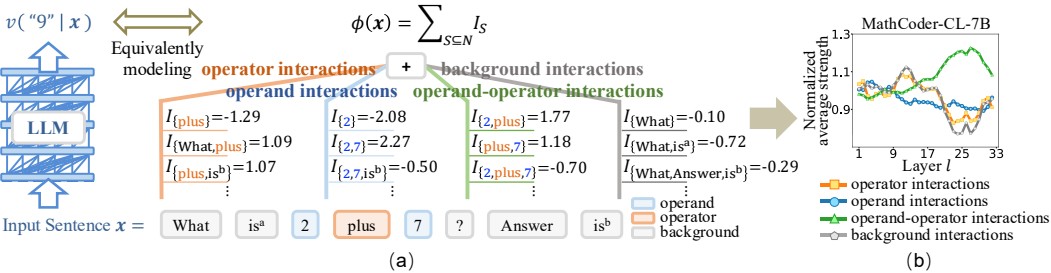

Figure 1: Illustration of using interactions to explain the internal mechanism of LLMs for arithmetic reasoning. (a) Given an arithmetic question $x$ consisting of $n$ words, let $N = \{x_1, x_2, \ldots, x_n\}$, where each $x_i$ represents a single word in the input question. We use a logic model based on interactions, i.e., $\phi(x) = \sum_{S \subseteq N} I_S$ to explain the detailed inference patterns encoded by LLMs. Note that superscripts a and b are not input to the LLM, but are used to distinguish two words of "*is*" in two positions. (b) Based on interactions, we investigate how an LLM encodes different types of interactions during forward propagation when solving arithmetic problems.

words in $x$, i.e., $N = \{x_1, x_2, \ldots, x_n\}$. This means that the output score of the LLM on any input sentence can be represented as the sum of the effects of all interactions.

• More crucially, no matter how we mask[3] input words in $x$, the output score of an LLM on the masked sentence $x^{\text{masked}}$ is equal to the output of the logical model, that is, $v(x^{\text{masked}}) = \phi(x^{\text{masked}}) = \sum_{S \subseteq N} \mathbb{1}(x^{\text{masked}} \text{ triggers } S) \cdot I_S$. This means that the output score of an LLM on any masked sentence can be represented as the sum of the effects of all triggered interactions. For example, if we mask the input word "*7*" in Figure 1(a) and obtain the masked sentence "*What is 2 plus [pad_token]? Answer is*," then all interactions that contain "*7*" (e.g., $S = \{2, plus, 7\}$) will not be triggered, meaning that $\mathbb{1}(x^{\text{masked}} \text{ triggers } S) \cdot I_S = 0$. As a result, these non-triggered interactions have no effects on the network output. Please see Theorem 1 for details.

In this way, interactions can be considered as inference patterns encoded by an LLM. Therefore, we quantify how LLMs encode different interactions during forward propagation. We find that different LLMs have preferences for encoding certain interactions. For example, as Figure 2 shows, when computing features closer to the output layer, the Llemma-7B [Azerbayev et al., 2023] model tends to encode interactions that contain both operands and operators, while the OPT-1.3B [Zhang et al., 2022b] model tends to encode interactions that only contain background words.

Inspired by the above observations, we further define four types of interactions encoded by the LLM for arithmetic reasoning, including (1) operand interactions, (2) operator interactions, (3) operand-operator interactions, and (4) background interactions, as shown in Figure 1. Furthermore, we propose a new metric to quantify the focus of LLMs on different types of interactions and discover the following insights.

• *Insight 1*: **The internal mechanism of LLMs for solving simple one-operator arithmetic problems is their capability to encode[2] operand-operator interactions and high-order interactions from input samples.** For simple one-operator arithmetic problems, we observe that LLMs with strong performance tend to increase their focus on operand-operator interactions and high-order interactions, while reducing their focus on operator interactions and background interactions. A high-order interaction contains a large number of input words, while a low-order interaction contains a small number of input words. In contrast, LLMs with weak performance tend to focus on background interactions and extremely low-order interactions.

• *Insight 2*: **The internal mechanism of LLMs for solving relatively complex two-operator arithmetic problems is their capability to encode[2] operator interactions and operand interactions from input samples.** As an LLM learns to solve two-operator arithmetic problems, we observe that it consistently focuses more on operator interactions and gradually increases its focus on operand interactions during the final stages of training.

---

[3]It is common to use a specific token or embedding to mask input variables of a DNN, but there are still no unified masking strategies. Please see Appendix A for an introduction to masking strategies.

- *Insight 3*: **We explain the task-specific nature of the LoRA method from the perspective of interactions.** As an LLM with strong one-operator arithmetic capabilities is fine-tuned using LoRA to solve relatively complex two-operator arithmetic problems, its capability to solve simpler one-operator problems declines. This phenomena is consistent with the task-specific nature of LoRA [Hu et al., 2022]. From the perspective of interactions, we observe that the LLM reduces its focus on operand-operator interactions and high-order interactions, which helps explain the underlying mechanism behind the task-specific nature of LoRA in arithmetic reasoning tasks.

## 2 Related Work

**Explaining arithmetic reasoning in LLMs.** Some studies identified and analyzed key neurons to explain arithmetic reasoning in LLMs. Yu and Ananiadou [2024] identified an internal logic chain. Rai and Yao [2024] investigated neuron activations and identified reasoning-related neurons. Some studies identified key components by changing the inputs of the LLM to perturb activations and then measuring the changes in the output logits. Hanna et al. [2023] explained the capability of the GPT2-small model on the "greater than" task by identifying a specific circuit. Zhang et al. [2024] further selectively fine-tuned the key components to boost the LLM's arithmetic performance. Stolfo et al. [2023] used a causal mediation analysis framework to provide a mechanistic explanation of how LLMs solve arithmetic problems. Wu et al. [2023] proposed a method to scale causal analysis of language models to billions of parameters. However, previous studies failed to provide strong mathematical support for their explanations. In contrast, in this paper, we use interactions to explain how LLMs solve arithmetic tasks, which has been proven to be a faithful explanation.

**Using game-theoretic interactions to explain DNNs.** Ren et al. [2023a] proposed using interactions to explain DNNs and provided a series of theoretical guarantees to ensure the faithfulness of this explanation. A series of studies further explored using interactions to explain the representational power of DNNs, including adversarial robustness [Wang et al., 2021], adversarial transferability [Wang et al., 2020], and the overfitting problem [Ren et al., 2023b, Zhou et al., 2023]. Additionally, Deng et al. [2021] demonstrated that DNNs struggle to encode mid-order interactions. Ren et al. [2024] discovered that DNNs encode interactions of different orders in two distinct phases. Some other studies used interactions to analyze common mechanisms across various deep-learning methods. Deng et al. [2024] showed that the core mechanism of fourteen attribution methods can be explained as a redistribution of interactions. Zhang et al. [2022a] found that the twelve previous methods for improving transferability all reduce interactions between local adversarial perturbations. However, due to terabytes of data and billions of parameters in LLMs, as well as the inherent complexity of arithmetic reasoning tasks, whether interactions can be used to explain the arithmetic reasoning of LLMs while ensuring the faithfulness of the explanation remains to be verified.

## 3 Using interactions to explain inference patterns encoded by LLMs

### 3.1 Preliminaries: disentangling the network output using interactions

Given a DNN and an input sample $x$ with $n$ variables indexed by $N = \{x_1, x_2, \ldots, x_n\}$, we can obtain $2^n$ masked[3] sentences $\{x_T \mid T \subseteq N\}$ by masking[3] each of $n$ variables. Specifically, $x_T$ denotes the input sentence when we keep variables in $T$ unchanged and mask[3] variables in $N \backslash T$. $x_\emptyset$ denotes the input sentence when all variables in $x$ are masked[3]. Ren et al. [2021b] have proven that there exists a surrogate logical model $\phi(\cdot)$ that can well predict/fit the scalar output of the DNN for all $2^n$ different masked sentences, as Theorem 1 shows.

$$\forall T \subseteq N, \phi(x_T) = \phi(x_\emptyset) + \sum_{S \subseteq N, S \neq \emptyset} \mathbb{1}(S|x_T) \cdot I_S, \tag{1}$$

where the trigger function $\mathbb{1}(S|x_T)$ represents an **AND** relationship among the input variables in a set $S \subseteq N$. That is, if all variables in $S$ are present in $T$, then $\mathbb{1}(S|x_T) = 1$. Otherwise, if any variable in $S$ is masked[3], then $\mathbb{1}(S|x_T) = 0$. Each interaction $S$ has a scalar weight $I_S$.

**Theorem 1.** *(Universal matching property, as proven by Ren et al. [2023a]). When the scalar weight $I_S$ in the logical model $\phi(\cdot)$ is defined as follows,*

$$\forall S \subseteq N, S \neq \emptyset, I_S = \sum_{S' \subseteq S} (-1)^{|S|-|S'|} \cdot v(x_{S'}), \tag{2}$$

*then we have $\forall T \subseteq N$, the output score of the DNN $v(\boldsymbol{x}_T) = \phi(\boldsymbol{x}_T)$.*

The universal matching property guarantees the theoretical faithfulness of using interactions to explain the DNN. That is, the interaction truly reflects the inference logic of the DNN. Besides, the interaction has been proven to serve as the basis for many game-theoretic metrics, further ensuring its theoretical faithfulness. Please see Appendix C for more details about game-theoretic metrics.

### 3.2 Explaining the LLM using interactions

Although interactions have been widely used to explain traditional DNNs, the following two challenges remain in using interactions to explain the forward propagation process of the LLM.

**(1) How to quantify interactions encoded by an LLM in intermediate layers.** Given an LLM with $L$ layers[4], we set the output score $v^{(l)}(\boldsymbol{x}_T)$ at layer $l$ as follows.

$$\forall T \subseteq N, v^{(l)}(\boldsymbol{x}_T) = cos(f^{(l)}(\boldsymbol{x}_T), f^{(l)}(\boldsymbol{x}_N)) = \frac{(f^{(l)}(\boldsymbol{x}_T))^\top f^{(l)}(\boldsymbol{x}_N)}{\|f^{(l)}(\boldsymbol{x}_T)\|_2 \|f^{(l)}(\boldsymbol{x}_N)\|_2}, \tag{3}$$

where $f^{(l)} \in \mathbb{R}^d$ denotes the embedding of the last input token.[5] The score $v^{(l)}(\boldsymbol{x}_T)$ measures the cosine similarity between $f^{(l)}(\boldsymbol{x}_T)$ and $f^{(l)}(\boldsymbol{x}_N)$. If the value of $v^{(l)}(\boldsymbol{x}_T)$ is large, then it indicates that the feature $f^{(l)}(\boldsymbol{x}_T)$ of the masked sentence is similar with the feature $f^{(l)}(\boldsymbol{x}_N)$ of the original sentence. This means that input variables in $T$ have great impact on the feature of the $l$-th layer. Thus, in Equation (2), the interaction computed based on the score can be considered as an inference pattern encoded by the features at the $l$-th layer. Note that we use $f^{(l)}(\boldsymbol{x}_T) - f^{(l)}(\boldsymbol{x}_\emptyset)$ to replace $f^{(l)}(\boldsymbol{x}_T)$, where $f^{(l)}(\boldsymbol{x}_\emptyset)$ denotes the embedding when we mask[3] all input variables in $\boldsymbol{x}$. This shifting operation is used to ensure that $f^{(l)}(\boldsymbol{x}_\emptyset) = \boldsymbol{0}$.

**(2) Why we use words instead of tokens as input variables.** It is because the tokenizers of different LLMs adopt different strategies for encoding numbers. For example, the Llama-2-7B [Touvron et al., 2023] model divides the word "*15*" into two tokens "*1*" and "*5*," while the OPT-1.3B [Zhang et al., 2022b] model obtains a single token "*15*." To ensure a fair comparison of interactions encoded by different LLMs, we treat all tokens within a word as a single input variable. When generating the masked[3] sentence $\boldsymbol{x}_T$, we use a specific padding token (see Appendix B for details) to mask all tokens that belong to the words in $N \setminus T$.

Based on the above two settings, we quantify interactions in LLMs when solving arithmetic problems. Figure 2 shows the top 10 interactions (i.e., those interactions with largest absolute values) encoded[2] by the Llemma-7B model [Azerbayev et al., 2023] and the OPT-1.3B model [Zhang et al., 2022b] when solving a one-operator problem "*How much is 4 times 2? Answer is.*" The Llemma-7B model predicts the correct answer "*8*," while the OPT-1.3B model predicts the incorrect answer "*4*." We observe that the Llemma-7B model mainly focuses on interactions that contain only background words in an early (5th) layer[4], and mainly focuses on interactions containing both operands and operators in a late (30th) layer. In comparison, the OPT-1.3B model mainly focuses on interactions that contain only background words in a late (24th) layer, which might be the reason why the OPT-1.3B model answers incorrectly.

To better observe the changing trends of different interactions encoded by an LLM during forward propagation, Figure 2 reports the curves of interactions encoded by LLMs when computing features. To ensure a fair comparison of interactions at different layers, we compute the normalized strength of interactions at different layers, that is, $\frac{|I_S^{(l)}|}{Z^{(l)}}$, where $Z^{(l)} = \mathbb{E}_{S \subseteq N}|I_S^{(l)}|$. As Figure 2 shows, the Llemma-7B model enhances its focus on interactions containing both operands and operators (e.g., $S = \{is^a, 4, times, 2\}$ ) during forward propagation. In comparison, the OPT-1.3B model does not show a preference for any interactions in the middle layers and shifts its focus to interactions that contain only background words (e.g., $S = \{is^b\}$) in the very few layers close to the output layer. Therefore, interactions provide us with a new perspective to explain the internal mechanism of LLMs for arithmetic reasoning.

---

[4]Please see Appendix D for details about network architectures and chosen layers.

[5]We follow Wendler et al. [2024] to focus on the embedding of the last input token at each layer $l$, as the last input token captures the information from the entire sentence. We also conduct experiments to verify that the mid-layer features of other tokens remain nearly unchanged when the input is masked. Please see Appendix E for details. To this end, we only use the embedding of the last input token to compute $v^{(l)}(\boldsymbol{x}_T)$.

Given a simple one-operator arithmetic problem as input: "How much is[a] 4 times 2? Answer is[b]"

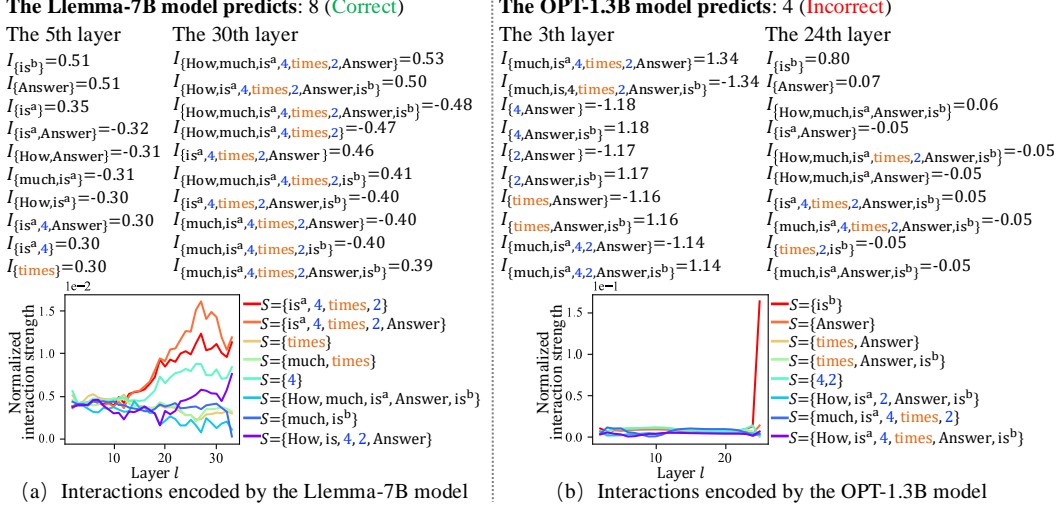

**The Llemma-7B model predicts**: 8 (Correct)

The 5th layer

$I_{\{is^b\}}=0.51$
$I_{\{Answer\}}=0.51$
$I_{\{is^a\}}=0.35$
$I_{\{is^a,Answer\}}=-0.32$
$I_{\{How,Answer\}}=-0.31$
$I_{\{much,is^a\}}=-0.31$
$I_{\{How,is^a\}}=-0.30$
$I_{\{is^a,4,Answer\}}=0.30$
$I_{\{is^a,4\}}=0.30$
$I_{\{times\}}=0.30$

The 30th layer

$I_{\{How,much,is^a,4,times,2,Answer\}}=0.53$
$I_{\{How,is^a,4,times,2,Answer,is^b\}}=0.50$
$I_{\{How,much,is^a,4,times,2,Answer,is^b\}}=-0.48$
$I_{\{How,much,is^a,4,times,2\}}=-0.47$
$I_{\{is^a,4,times,2,Answer\}}=0.46$
$I_{\{How,much,is^a,4,times,2,is^b\}}=0.41$
$I_{\{is^a,4,times,2,Answer,is^b\}}=-0.40$
$I_{\{much,is^a,4,times,2,Answer\}}=-0.40$
$I_{\{much,is^a,4,times,2,is^b\}}=-0.40$
$I_{\{much,is^a,4,times,2,Answer,is^b\}}=0.39$

**The OPT-1.3B model predicts**: 4 (Incorrect)

The 3th layer

$I_{\{much,is^a,4,times,2,Answer\}}=1.34$
$I_{\{much,is,4,times,2,Answer,is^b\}}=-1.34$
$I_{\{4,Answer\}}=-1.18$
$I_{\{4,Answer,is^b\}}=1.18$
$I_{\{2,Answer\}}=-1.17$
$I_{\{2,Answer,is^b\}}=1.17$
$I_{\{times,Answer\}}=-1.16$
$I_{\{times,Answer,is^b\}}=1.16$
$I_{\{much,is^a,4,2,Answer\}}=-1.14$
$I_{\{much,is^a,4,2,Answer,is^b\}}=1.14$

The 24th layer

$I_{\{is^b\}}=0.80$
$I_{\{Answer\}}=0.07$
$I_{\{How,much,is^a,Answer,is^b\}}=0.06$
$I_{\{is^a,Answer\}}=-0.05$
$I_{\{How,much,is^a,times,2,Answer,is^b\}}=-0.05$
$I_{\{How,much,is^a,Answer\}}=-0.05$
$I_{\{is^a,4,times,2,Answer,is^b\}}=0.05$
$I_{\{much,is^a,4,times,2,Answer,is^b\}}=-0.05$
$I_{\{times,2,is^b\}}=-0.05$
$I_{\{much,is^a,Answer,is^b\}}=-0.05$

(a) Interactions encoded by the Llemma-7B model

(b) Interactions encoded by the OPT-1.3B model

Figure 2: Visualization of top 10 interactions encoded by LLMs when computing features in different layers. Results show that the Llemma-7B model mainly focuses on interactions containing both operands (in blue) and operators (in orange) in a late layer (i.e., the 30th layer). In comparison, the OPT-1.3B model mainly focuses on interactions that contain only background words in a late layer (i.e., the 24th layer), which may be the reason why the OPT-1.3B model answers incorrectly. Note that superscripts a and b are not input to the LLM, but are used to distinguish two words of "*is*."

## 3.3 Defining and quantifying different types of interactions

Inspired by the above observations obtained in Section 3.2, we classify the variables in an input arithmetic query $\boldsymbol{x}$ into the following three categories: operand variables $\boldsymbol{x}^{\text{opd}}$, operator variables $\boldsymbol{x}^{\text{opr}}$ and background variables $\boldsymbol{x}^{\text{bg}}$. Thus, all $2^n$ interactions can be classified as the following four types to describe the inference patterns encoded by LLMs for arithmetic reasoning.

*Operand interactions.* If an interaction $S$ contains at least one operand variable but no operator variables, we regard $S$ as an operand interaction. Let $\Omega^{\text{opd}}$ denote the set of all operand interactions.

$$\Omega^{\text{opd}} = \{S \mid \exists \boldsymbol{x}^{\text{opd}} \in S \wedge \nexists \boldsymbol{x}^{\text{opr}} \in S\}. \tag{4}$$

*Operator interactions.* If an interaction $S$ contains at least one operator variable but no operand variables, we regard $S$ as an operator interaction. Let $\Omega^{\text{opr}}$ denote the set of all operator interactions.

$$\Omega^{\text{opr}} = \{S \mid \exists \boldsymbol{x}^{\text{opr}} \in S \wedge \nexists \boldsymbol{x}^{\text{opd}} \in S\}. \tag{5}$$

*Operand-operator interactions.* If an interaction $S$ contains at least one operand variable and at least one operator variable, we regard $S$ as an operand-operator interaction. Let $\Omega^{\text{opd-opr}}$ denote the set of all operand-operator interactions.

$$\Omega^{\text{opd-opr}} = \{S \mid \exists \boldsymbol{x}^{\text{opd}} \in S \wedge \exists \boldsymbol{x}^{\text{opr}} \in S\}. \tag{6}$$

*Background interactions.* If an interaction $S$ contains neither operand nor operator variables, we regard $S$ as a background interaction. Let $\Omega^{\text{bg}}$ denote the set of all background interactions.

$$\Omega^{\text{bg}} = \{S \mid \nexists \boldsymbol{x}^{\text{opd}} \in S \wedge \nexists \boldsymbol{x}^{\text{opr}} \in S\}. \tag{7}$$

Note that the union of the four sets above is the complete set of $2^n$ interactions. **According to Theorem 1, the sum of numerical effects of these four types of interactions can accurately fit the output scores of the LLM, thereby ensuring the faithfulness of the analysis.** We design the following metric to quantify how an LLM focuses on a specific type of interaction $\Omega^{type} \in \{\Omega^{\text{opd}}, \Omega^{\text{opr}}, \Omega^{\text{opd-opr}}, \Omega^{\text{bg}}\}$.

**Definition 1.** *(Focality on a specific type of interaction). Let $R^{(l)}(\Omega^{type})$ denote the focality on a specific type of interaction $\Omega^{type}$ at layer l. It is computed as follows,*

$$R^{(l)}(\Omega^{type}) = \frac{\mathbb{E}_{S \in \Omega^{type}} |I_S^{(l)}|}{Z^{(l)}}, \tag{8}$$

where $Z^{(l)} = \mathbb{E}_{S \subseteq N}|I_S^{(l)}|$ is a normalization term used to ensure a fair comparison of the interaction effects across different layers.

In Equation 8, a higher $R^{(l)}(\Omega^{type})$ value suggests that the LLM focuses more on this type of interaction $\Omega^{type}$ when computing features at layer $l$. If $R^{(l)}(\Omega^{type}) = 1$, that means the strength of this type of interaction $\Omega^{type}$ encoded by the LLM is equal to the average strength of all interactions encoded by the LLM, which reveals that the LLM does not show a preference for this type of interaction. If $R^{(l)}(\Omega^{type}) > 1$, that means the strength of this type of interaction $\Omega^{type}$ exceeds the average, indicating that the LLM exhibits a preference for this type of interaction.

We also quantify interactions of different orders to measure the representation complexity of an LLM. The order $m$ of an interaction $S$ is defined as the number of variables in $S$, that is, $m = |S|$. We design the following metric to quantify how an LLM focuses on $m$-order interactions.

**Definition 2.** *(Focality on interactions of a specific order). Let $\kappa_m^{(l)}$ denote the focality on $m$-order interactions at layer $l$. It is computed as follows,*

$$\forall m \in \{1, 2, \ldots, n\}, \kappa_m^{(l)} = \frac{\mathbb{E}_{|S|=m}|I_S^{(l)}|}{Z^{(l)}}. \tag{9}$$

If $\kappa_m^{(l)}$ has a larger value when $m$ is higher, it indicates that the LLM encodes interactions of greater complexity when computing features at layer $l$. If $\kappa_m^{(l)}$ has a larger value when $m$ is lower, it indicates that the LLM encodes interactions of lower complexity when computing features at layer $l$.

## 4 Comparative studies

In this section, we conduct comparative studies to analyze the internal mechanism of LLMs for arithmetic reasoning (see Section 4.1). We also fine-tune an LLM to improve its capability to solve arithmetic problems and explore how the LLM encodes different types of interactions during the training process (see Section 4.2).

**LLMs.** We use interactions to analyze seven LLMs for arithmetic reasoning, including the OPT-1.3B [Zhang et al., 2022b] model, the GPT-J-6B [Wang and Komatsuzaki, 2021] model, the Llama-2-7B [Touvron et al., 2023] model, the Llemma-7B [Azerbayev et al., 2023] model, the MathCoder-L-7B [Wang et al., 2023] model, the MathCoder-CL-7B [Wang et al., 2023] model, and the CodeLlama-13B [Roziere et al., 2023] model. Appendix B shows how to mask words for these LLMs.

**Data.** We follow Karpas et al. [2022], Razeghi et al. [2022], Stolfo et al. [2023] to conduct experiments on a set of arithmetic problems hand-crafted by humans, including 6 templates for one-operator two-operand queries and 29 templates for two-operator three-operand queries. For example, "*The sum of $n_1$ and $n_2$ is*" and "*What is the ratio between $n_1$ minus $n_2$ and $n_3$? The answer is.*" Each template for one-operator queries includes all four arithmetic operators, i.e., $\{+, -, \times, \div\}$, and each template for two-operator queries corresponds to a unique com-

Table 1: Overall accuracy (%) of different LLMs on arithmetic queries.

| Model | 1-opr | 2-opr |
|---|---|---|
| OPT-1.3B | 3.2 | 1.7 |
| GPT-J-6B | 14.7 | 5.8 |
| Llama-2-7B | 65.1 | 10.1 |
| Llemma-7B | 75.1 | 15.3 |
| MathCoder-L-7B | 74.0 | 8.2 |
| MathCoder-CL-7B | 62.6 | 9.3 |
| CodeLlama-13B | 71.1 | 15.0 |
| OPT-1.3B Fine-tuned | 83.6 | 69.7 |

bination of two operators. Please see Appendix F for details of templates. For each template of one-operator queries, we generate 20 prompts by randomly sampling operands $(n_1, n_2)$,[6] and for each template of two-operator queries, we generate 5 prompts following the same procedure. Table 1 shows the overall accuracy of different LLMs on one-operator queries and two-operator queries. Please see Appendix G for details of the accuracy tests. We observe that the Llama-2-7B model, the Llemma-7B model, the MathCoder-L-7B model, the MathCoder-CL-7B model, and the CodeLlama-13B model perform well on one-operator queries, while the OPT-1.3B model and the GPT-J-6B model perform relatively poorly. However, for two-operator queries, all seven LLMs perform poorly.

---

[6]We sample operands from $\{1, 2, \ldots, 9\}$ since some LLMs tokenize each digit as an independent token, such as the Llama-2-7B model.

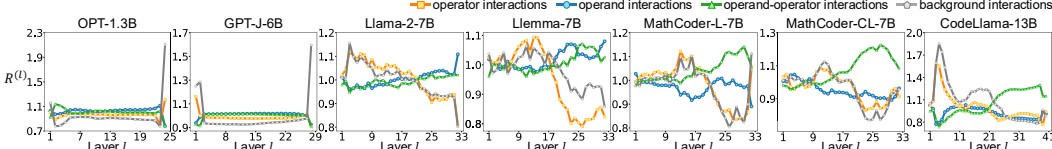

Figure 3: Focality across different types of interactions $R^{(l)}$ encoded by LLMs during forward propagation. Each curve in the figure is averaged over various one-operator arithmetic queries.

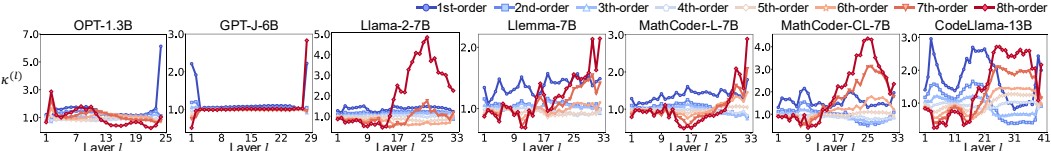

Figure 4: Focality across interactions of different orders $\kappa^{(l)}$ encoded by LLMs during forward propagation. Each curve in the figure is averaged over various one-operator arithmetic queries.

### 4.1 Exploring the internal mechanism of LLMs for solving arithmetic problems

In this subsection, we analyze how LLMs focus on different types of interactions during forward propagation and obtain the following insights.

**Insight 1: The internal mechanism of LLMs for solving simple one-operator arithmetic problems is their capability to encode operand-operator interactions and high-order interactions.**
Figure 3 reports the focality $R^{(l)}$ on different types of interactions encoded by LLMs during forward propagation. The results are averaged over all one-operator queries. We observe that in LLMs with strong one-operator arithmetic capabilities (the Llama-2-7B model, the Llemma-7B model, the MathCoder-L-7B model, the MathCoder-CL-7B model, and the CodeLlama-13B model), the focality on operand-operator interactions $R^{(l)}(\Omega^{\text{opd-opr}})$ tends to increase and exceeds 1.0 in the later layers, while the focality on operator interactions $R^{(l)}(\Omega^{\text{opr}})$ and the focality on background interactions $R^{(l)}(\Omega^{\text{bg}})$ tend to fall below 1.0. We notice that in the MathCoder-L-7B model, the focality on operator interactions and the focality on background interactions increase and exceed 1.0 in the very few layers close to the output layer. However, the encoding of operand-operator interactions in the later layers remains sufficient to support its arithmetic reasoning. This suggests that solving one-operator arithmetic problems needs LLMs to increase their focus on operand-operator interactions, and reduce their focus on operator interactions and background interactions in the later layers. Note that in the Llama-2-7B model, the Llemma-7B model, and the MathCoder-L-7B model, the focality on operand interactions $R^{(l)}(\Omega^{\text{opd}})$ also tends to increase and exceeds 1.0 in the later layers, which may contribute to their strong performance on one-operator arithmetic tasks.

In comparison, in LLMs with weak one-operator arithmetic capabilities (the OPT-1.3B model and the GPT-J-6B model), the focality on different types of interactions is similar and remains around 1.0 in the middle layers, while the focality on background interactions $R^{(l)}(\Omega^{\text{bg}})$ suddenly increases in the very few layers close to the output layer. This suggests that these LLMs do not exhibit a clear preference for any specific type of interaction in the middle layers, but subsequently shift their focus to background interactions in the very few layers close to the output layer, which weakens their performance on one-operator arithmetic tasks.

Figure 4 reports the focality $\kappa^{(l)}$ on interactions of different orders encoded by LLMs during forward propagation. The results are averaged over queries from a single one-operator template.[7] We observe that in LLMs with strong one-operator arithmetic capabilities (the Llama-2-7B model, the Llemma-7B model, the MathCoder-L-7B model, the MathCoder-CL-7B model, and the CodeLlama-13B model), the focality on high-order interactions tends to increase and exceeds 1.0 in the later layers. This suggests that solving one-operator arithmetic problems needs LLMs to increase their focus on high-order interactions in the later layers. In comparison, in LLMs with weak one-operator arithmetic capabilities (the OPT-1.3B model and the GPT-J-6B model), the focality on high-order interactions in the middle layers is around 1.0, while in the very few layers close to the output layer, the focality on

---

[7]As different templates correspond to different maximum orders, i.e., the number of input words. Please see Appendix I for more results from other templates, which lead to the same conclusion.

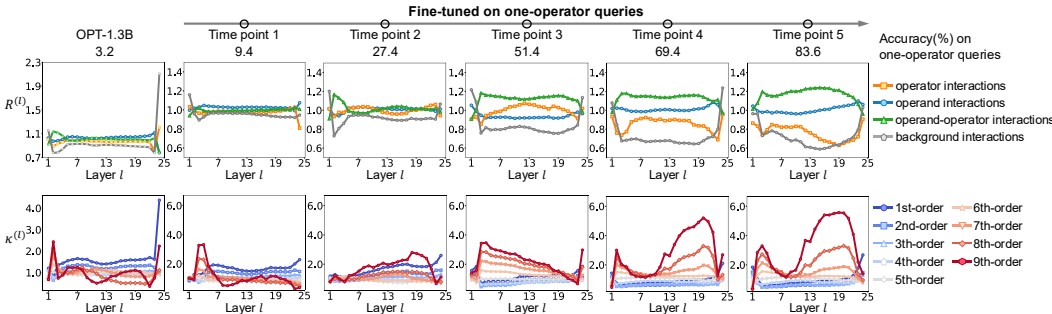

Figure 5: Visualizing the dynamic changes in different types and orders of interactions encoded by the OPT-1.3B model during the training process on one-operator queries. Each curve is averaged over various one-operator queries. Results show that as accuracy improves, the model gradually increases its focus on operand-operator interactions (top) and high-order interactions (bottom).

the extremely low-oder interactions suddenly increases. This suggests these LLMs exhibit insufficient encoding of higher-order interactions in the middle layers and tend to focus on extremely low-order interactions, which weakens their performance on one-operator arithmetic tasks.

## 4.2 Dynamic encoding of different types of interactions during the training process

We further explore how an LLM learns to solve arithmetic problems. That is, we investigate how an LLM encodes different types of interactions when trained on arithmetic problem data. To this end, we use the LoRA method [Hu et al., 2022] to fine-tune the OPT-1.3B model on arithmetic queries in the following three different ways.[8] (1) We fine-tune the OPT-1.3B model on one-operator queries, improving its accuracy on one-operator queries from 3.2% to 83.6%. This version is termed the *OPT-1.3B-One* model. (2) We fine-tune the OPT-1.3B model on two-operator queries, improving its accuracy on two-operator queries from 1.7% to 69.7%. (3) Building upon the *OPT-1.3B-One* model, we further train it on two-operator queries. We analyze the dynamic encoding of different types of interactions during the above three training processes and obtain the following insights.

Figure 5 reports the dynamic changes in different types of interactions encoded by the OPT-1.3B model during training on one-operator queries. The results of $R^{(l)}$ are averaged over all one-operator queries, and the results of $\kappa^{(l)}$ are averaged over queries from a single one-operator template.[7] We observe that as the accuracy of the OPT-1.3B model on one-operator queries improves, the focality on operand-operator interactions $R^{(l)}(\Omega^{\text{opd-opr}})$ and the focality on high-order interactions gradually increase, while the focality on operator interactions $R^{(l)}(\Omega^{\text{opr}})$ and the focality on background interactions $R^{(l)}(\Omega^{\text{bg}})$ gradually decrease. **The results validate our Insight 1.**

We also observe that the OPT-1.3B model consistently enhances its focus on background interactions in the very few layers close to the output layer during the training process. This may be due to the inherent preference of the OPT-1.3B model, which tends to enhance its focus on background interactions in the very few layers close to the output layer, as Figure 3 shows.

**Insight 2: The internal mechanism of LLMs for solving relatively complex two-operator arithmetic problems is their capability to encode operator interactions and operand interactions.** Figure 6 reports the dynamic changes in different interactions encoded by the OPT-1.3B model during training on two-operator queries. The results are averaged over all two-operator queries. We observe that the focality on operator interactions $R^{(l)}(\Omega^{\text{opr}})$ remains consistently high throughout the training process. As the accuracy of the OPT-1.3B model on two-operator queries improves, the focality on operand interactions $R^{(l)}(\Omega^{\text{opd}})$ gradually increases in the later layers during the final stage of training. This suggests that solving two-operator arithmetic problems requires an LLM to focus more on operator interactions and operand interactions. We notice that the OPT-1.3B model consistently focuses on background interactions, which may be due to its inherent preference. Notably, the focality on operand-operator interactions gradually increases in the very few layers close to the output layer, which may contribute to the model's strong performance on two-operator arithmetic tasks.

---

[8]Please see Appendix H for details about model training.

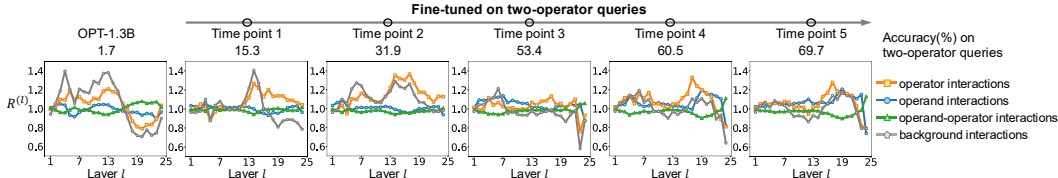

Figure 6: Visualizing the dynamic changes in different types of interactions encoded by the OPT-1.3B model during the training process on two-operator queries. Each curve is averaged over various two-operator queries. Results show that the OPT-1.3B model consistently focuses more on operator interactions and gradually increase its focus on operand interactions during the final stage of training.

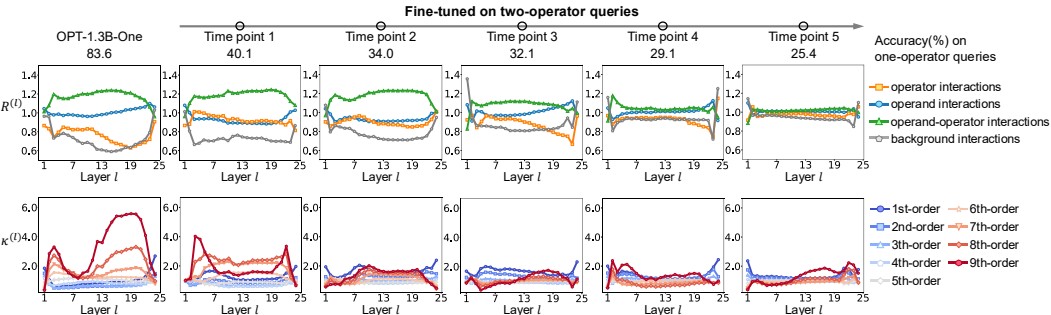

Figure 7: Visualizing the dynamic changes in different types and orders of interactions encoded by the *OPT-1.3B-One* model during the training process on two-operator queries. Each curve is averaged over various one-operator queries. Results show that as accuracy declines, the model reduces its focus on operand-operator interactions (top) and high-order interactions (bottom).

**Insight 3: We explain the task-specific nature of the LoRA method from the perspective of interactions.** Figure 7 reports the dynamic changes in different types of interactions encoded by the *OPT-1.3B-One* model during training on two-operator queries. The results of $R^{(l)}$ are averaged over all one-operator queries, and the results of $\kappa^{(l)}$ are averaged over queries from a single one-operator template.[7] We observe that the accuracy of the *OPT-1.3B-One* model on one-operator queries declines, which means that the *OPT-1.3B-One* model gradually forgets how to solve simpler arithmetic problems while learning more complex knowledge. Meanwhile, the accuracy on two-operator queries steadily improves during training (please see Appendix M for details). This phenomena is consistent with the task-specific nature of LoRA [Hu et al., 2022]. From the perspective of interactions, we observe that the focality on operand-operator interactions $R^{(l)}(\Omega^{\mathrm{opd\text{-}opr}})$ and the focality on high-order interactions gradually decrease during the training process, with most values tending to stay around 1.0. The results help explain the underlying mechanism behind the task-specific nature of LoRA in arithmetic reasoning tasks and validate our Insight 1.

## 5   Conclusion and discussions

In this paper, we use interactions to provide a deep understanding of the internal mechanism of LLMs for arithmetic reasoning. Through comparison studies of different types and orders of interactions encoded by LLMs during forward propagation, we find that the internal mechanism of LLMs for solving simple one-operator arithmetic problems is their capability to encode operand-operator interactions and high-order interactions. We further fine-tune an LLM to explore how an LLM encodes different types and orders of interactions when learning to solve arithmetic problems. We find that the internal mechanism of LLMs for solving relatively complex two-operator arithmetic problems is their capability to encode operator interactions and operand interactions. We also explain the task-specific nature of the LoRA method from the perspective of interactions.

**Limitations.** As the number of input variables increases significantly, the computational cost of interaction analysis indeed rises. To address this issue, we can analyze informative variables while treating uninformative variables as background or adopt some other methods (see Appendix L). On the other hand, we have only studied simple arithmetic problems and have not yet extended our research to more complex math word problems. In the future, we will work on this.

## Acknowledgment

This work is partially supported by the National Nature Science Foundation of China (No.62376199,62206170).

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

## A  Strategies of masking input variables

In the research of attribution methods, it is common to use a specific token or embedding to mask input variables of a DNN [Lundberg, 2017, Ancona et al., 2019, Fong et al., 2019], and use changes in network outputs on the masked samples to estimate attributions of input variables. However, each method has certain limitations. The mean baseline value [Dabkowski and Gal, 2017], i.e., setting the baseline value for each input variable to its mean across all samples, introduces additional signals, e.g., grey dots in images. Similarly, the zero baseline value [Ancona et al., 2019, Sundararajan et al., 2017], i.e., setting baseline values for all input variables to zero, would also introduce additional signals to the input, such as black dots.

## B  Details about how to mask input words for different LLMs

In this paper, we analyze seven LLMs for arithmetic reasoning, including the OPT-1.3B model, the GPT-J-6B model, the Llama-2-7B model, the CodeLlama-2-7B model, the MathCoder-L-7B model, the MathCoder-CL-7B model, and the Llemma-7B model. For the OPT-1.3B model, we use the "$$" token with the token $id = 2$ to mask the words in $N \backslash T$. For the GPT-J-6B model, we use the "$<|endoftext|>$" with the token $id = 50256$ to mask the words in $N \backslash T$. For the other five models, including the Llama-2-7B model, the CodeLlama-2-7B model, the MathCoder-L-7B model, the MathCoder-CL-7B model, and the Llemma-7B model, with Llama as their base model, we use the "$<unk>$" with the token $id = 0$ to mask the words in $N \backslash T$.

## C  Properties of interactions

The Harsanyi interaction $I_S$, i.e., the interaction in this paper, can explain the elementary mechanism of existing game-theoretic metrics [Ren et al., 2021a], including the *Shapley value* [Shapley, 1953], the *Shapley interaction index* [Grabisch and Roubens, 1999], and the *Shapley-Taylor interaction index* [Sundararajan et al., 2020].

(1) *Connection to the Shapley value* [Shapley, 1953]. Let $\phi(i)$ denote the Shapley value of an input variable $i$, given the input sample $x$. Then, the Shapley value $\phi(i)$ can be explained as the result of uniformly assigning attributions of each Harsanyi interaction to each involving variable $i$, i.e., $\phi(i) = \sum_{S \subseteq N \backslash \{i\}} \frac{1}{|S|+1} I_{S \cup \{i\}}$. This also proves that the Shapley value is a fair assignment of attributions from the perspective of the Harsanyi interaction.

(2) *Connection to the Shapley interaction index* [Grabisch and Roubens, 1999]. Given a subset of variables $T \subseteq N$ in an input sample $x$, the Shapley interaction index $I_T^{\text{Shapley}}$ can be represented as $I_T^{\text{Shapley}} = \sum_{S \subseteq N \backslash T} \frac{1}{|S|+1} I_{S \cup T}$. In other words, the index $I_T^{\text{Shapley}}$ can be explained as uniformly allocating $I_{S'}$ such that $S' = S \cup T$ to the compositional variables of $S'$, if we treat the coalition of variables in $T$ as a single variable.

(3) *Connection to the Shapley Taylor interaction index* [Sundararajan et al., 2020]. Given a subset of variables $T \subseteq N$ in an input sample $x$, the $k$-th order Shapley Taylor interaction index $I_T^{\text{Shapley-Taylor}}$ can be represented as a weighted sum of interaction effects, i.e., $I_T^{\text{Shapley-Taylor}} = I_T$ if $|T| < k$; $I_T^{\text{Shapley-Taylor}} = \sum_{S \subseteq N \backslash T} \binom{|S|+k}{k}^{-1} I_{S \cup T}$ if $|T| = k$; and $I_T^{\text{Shapley-Taylor}} = 0$ if $|T| > k$.

Given an input sample $x$, the Harsanyi interaction $I_S$ satisfies seven desirable axioms in game theory [Ren et al., 2021a], including the *efficiency, linearity, dummy, symmetry, anonymity, recursive and interaction* distribution axioms.

(1) *Efficiency axiom.* The output score of a model can be decomposed into interaction effects of different patterns, i.e., $v(x) = \sum_{S \subseteq N} I_S$.

(2) *Linearity axiom.* If we merge output scores of two models $w$ and $v$ as the output of model $u$, i.e., $\forall S \subseteq N, u(x_S) = v(x_S) + w(x_S)$, then their interaction effects $I_S^{(v)}$ and $I_S^{(w)}$ can also be merged as $\forall S \subseteq N, I_S^{(u)} = I_S^{(v)} + I_S^{(w)}$.

(3) *Dummy axiom.* If a variable $i \in N$ is a dummy variable, i.e., $\forall S \subseteq N \backslash \{i\}, v(x_{S \cup \{i\}}) = v(x_S) + v(x_{\{i\}})$, then it has no interaction with other variables, $\forall \emptyset \neq T \subseteq N \backslash \{i\}, I_{T \cup \{i\} = 0}$.

(4) *Symmetry axiom.* If input variables $i, j \in N$ cooperate with other variables in the same way, $\forall S \subseteq N \setminus \{i, j\}, v(\boldsymbol{x}_{S \cup \{i\}}) = v(\boldsymbol{x}_{S \cup \{j\}})$, then they have the same interaction effects with other variables, $\forall S \subseteq N \setminus \{i, j\}, I_{S \cup \{i\}} = I_{S \cup \{j\}}$.

(5) *Anonymity axiom.* For any permutations $\pi$ on $N$, we have $\forall S \subseteq N, I_S^{(v)} = I_{\pi S}^{(\pi v)}$, where $\pi S \triangleq \{\pi(i) \mid i \in S\}$, and the new model $\pi v$ is defined by $(\pi v)(\boldsymbol{x}_{\pi S}) = v(\boldsymbol{x}_S)$. This indicates that interaction effects are not changed by permutation.

(6) *Recursive axiom.* The interaction effects can be computed recursively. For $i \in N$ and $S \subseteq N \setminus \{i\}$, the interaction effect of the pattern $S \cup \{i\}$ is equal to the interaction effect of $S$ with the presence of $i$ minus the interaction effect of $S$ with the absence of $i$, i.e., $\forall S \subseteq N \setminus \{i\}, I_{S \cup \{i\}} = I_S^{(i \text{ is always present})} - I_S. I_S^{(i \text{ is always present})}$ denotes the interaction effect when the variable $i$ is always present as a constant context, i.e., $I_S^{(i \text{ is always present})} = \sum_{L \subseteq S} (-1)^{|S| - |L|} \cdot v(\boldsymbol{x}_{L \cup \{i\}})$.

(7) *Interaction distribution axiom.* This axiom characterizes how interactions are distributed for "interaction functions" [Sundararajan et al., 2020]. An interaction function $v_T$ parameterized by a subset of variables $T$ is defined as follows: $\forall S \subseteq N, v_T(\boldsymbol{x}_S) = c$, if $T \subseteq S$; otherwise, $v_T(\boldsymbol{x}_S) = 0$. The function $v_T$ models pure interaction among the variables in $T$ because only if all variables in $T$ are present the output value will be increased by $c$. The interactions encoded in the function $v_T$ satisfy $I_T = c$, and $\forall S \neq T, I_S = 0$.

# D   Details about network architectures and chosen layers for experiments in section 4

**OPT-1.3B model.** The OPT-1.3B model is composed of the following parts: one word embedding layer (namely *Embedding Layer*), one position embedding layer (namely *OPTLearnedPositionalEmbedding*), 24 OPT decoder modules (namely *OPTDecoderLayer*), and one linear output layer (namely *Linear Layer*). The architecture can be summarized as *Embedding Layer → OPTLearnedPositionalEmbedding → [OPTDecoderLayer]×24 → Linear Layer*. Each module of *OPTDecoderLayer* contains a self-attention mechanism layer (namely *OPTAttention*), an activation function (namely *ReLU*), a layer normalization operation (namely *LayerNorm*), two fully connected layers (namely *Linear*), and a final layer normalization operation (namely *LayerNorm*). In our experiments, we selected all 24 *OPTDecoderLayer* modules and conducted experiments based on the output features of the *LayerNorm* operation.

**GPT-J-6B model.** The GPT-J-6B model includes a single word embedding layer (namely *Embedding Layer*), followed sequentially by 28 Transformer blocks (namely *GPTJBlock*), culminating in an output layer normalization (namely *LayerNorm*) and a linear output layer (namely *Linear Layer*). This architecture can be succinctly described as: *Embedding Layer → [GPTJBlock]×28 → LayerNorm → Linear Layer*. Delving into the details of each module of *GPTJBlock*, it comprises three integral components, including a layer normalization (namely *LayerNorm*), a self-attention mechanism (namely *GPTJAttention*), and a feed-forward network (namely *GPTMLP*). In our experiments, we selected all 28 *GPTJBlock* modules and conducted experiments based on the output features of the *LayerNorm* operation.

**Other Llama-based models.** The other five models based on Llama include the Llama-2-7B model, the CodeLlama-2-7B model, the MathCoder-L-7B model, the MathCoder-CL-7B model, and the Llemma-7B model. These models share the same architecture, which is composed of various elements: a single word embedding layer (namely *Embedding Layer*), followed by a sequence of 32 *LlamaDecoderLayer*, an output layer normalization layer (namely *LlamaRMSNorm*), and culminating in a linear output layer (namely *Linear Layer*). This architecture can be summarized as *Embedding Layer → [LlamaDecoderLayer]×32 → LlamaRMSNorm → Linear Layer*. Delving into each module of *LlamaDecoderLayer*, it integrates multiple components, including a self-attention mechanism layer (namely *LlamaAttention*), a feed-forward network layer (namely *LlamaMLP*) encompassing several linear layers and activation functions, an input layer normalization (namely *Input LayerNorm*), and a post-attention layer normalization (namely *Post Attention LayerNorm*), the latter serving to normalize the output from the self-attention mechanism layer. In our experiments, we selected all 32 modules of *LlamaDecoderLayer* and conducted experiments based on the output features of the *LayerNorm* operation.

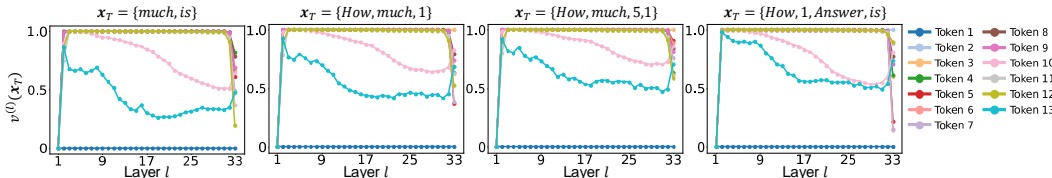

Figure 8: The visualization of $v^{(l)}$ at different token positions across various layers of CodeLlama for the prompt "How much is 5 plus 1? Answer is" under different masked inputs. Results show that except for the final tokens of the complete sentence (i.e., token 10 and token 13), the $v^{(l)}$ values of other tokens in the middle layers remain almost unchanged.

Table 2: Question templates for one-operator arithmetic queries.

| Type | Addition | Subtraction |
|------|----------|-------------|
| 1 | How much is $n_1$ plus $n_2$? Answer is | How much is $n_1$ minus $n_2$? Answer is |
| 2 | What is $n_1$ plus $n_2$? Answer is | What is $n_1$ minus $n_2$? Answer is |
| 3 | How much is the sum of $n_1$ and $n_2$? Answer is | How much is the difference between $n_1$ and $n_2$? Answer is |
| 4 | What is the sum of $n_1$ and $n_2$? Answer is | What is the difference between $n_1$ and $n_2$? Answer is |
| 5 | The sum of $n_1$ and $n_2$ is | The difference between $n_1$ and $n_2$ is |
| 6 | Given two numbers $n_1$ and $n_2$, the sum of them is | Given two numbers $n_1$ and $n_2$, the difference between them is |

| | Multiplication | Division |
|------|----------------|----------|
| 1 | How much is $n_1$ times $n_2$? Answer is | How much is $n_1$ over $n_2$? Answer is |
| 2 | What is $n_1$ times $n_2$? Answer is | What is $n_1$ over $n_2$? Answer is |
| 3 | What is the result of $n_1$ times $n_2$? Answer is | What is the result of $n_1$ over $n_2$? Answer is |
| 4 | How much is the product of $n_1$ and $n_2$? Answer is | How much is the ratio between $n_1$ and $n_2$? Answer is |
| 5 | The product of $n_1$ and $n_2$ is | The ratio of $n_1$ and $n_2$ is |
| 6 | Given two numbers $n_1$ and $n_2$, the product of them is | Given two numbers $n_1$ and $n_2$, the ratio between them is |

# E  The features of other tokens in the middle layers remain unchanged

Through experiments, we found that, except for the final token of the complete sentence, the $v^{(l)}$ values at other token positions in the middle layers remain almost unchanged. Figure 8 shows the $v^{(l)}$ values at different token positions across various layers of CodeLlama for the prompt "How much is 5 plus 1? Answer is" under different masked inputs $x_T$. The prompt consists of 13 tokens {``, `How`, `much`, `is`, `space`, `5`, `plus`, `space`, `1`, `?`, `answer`, `is`, `space`}. Except for token 10 (`?`) and token 13 (`space`), the $v^{(l)}$ values of other tokens in the middle layers remain almost unchanged.

# F  Prompt Templates

In Table 2 and 3, we report the question templates used as prompts for the model for one- and two-operator queries, respectively. For two-operator queries, we use one query template for each of the 29 possible two-operation combinations. To enable the model to output the answer directly, we appended "The answer is" at the end of each template.

# G  Performance of the LLMs

To more accurately and fairly evaluate the LLMs' capabilities to solve one-operator and two-operator arithmetic problems, we evaluated the accuracy of all LLMs on the one-operator and two-operator test sets used for fine-tuning in Appendix H.

For the OPT-1.3B and GPT-J-6B models, we treat each operand as a single token, while for other LLMs, each number is split into multiple tokens ("0", "1", "2", ..., "9") by the tokenizer.

Table 3: Question templates for two-operator arithmetic queries.

| Type | Formula | Format |
|------|---------|--------|
| 1 | $f = ((A + B) * C)$ | Sum A and B and multiply by C |
| 2 | $f = (A + B * C)$ | What is the sum of A and the product of B and C? |
| 3 | $f = ((A - B) * C)$ | What is the product of A minus B and C? |
| 4 | $f = (A/(B/C))$ | How much is A divided by the ratio between B and C? |
| 5 | $f = (A - B * C)$ | What is the difference between A and the product of B and C? |
| 6 | $f = (A * (B - C))$ | How much is A times the difference between B and C? |
| 7 | $f = ((A + B)/C)$ | What is the ratio between A plus B and C? |
| 8 | $f = (A - (B - C))$ | How much is A minus the diffrence between B and C? |
| 9 | $f = ((A - B)/C)$ | What is the ratio between A minus B and C? |
| 10 | $f = (A - B/C)$ | What is the difference between A and the ratio between B and C? |
| 11 | $f = (A/(B + C))$ | How much is A divided bu the sum of B and C? |
| 12 | $f = (A/(B - C))$ | How much is A divided by the difference between B and C? |
| 13 | $f = (A + B/C)$ | what is the sum of A and the ratio between B and C? |
| 14 | $f = (A * (B/C))$ | How much is A times the ratio between B and C? |
| 15 | $f = (A * B + C)$ | How much is the sum of A times B and C? |
| 16 | $f = (A * (B + C))$ | How much is A times the sum of B and C? |
| 17 | $f = (A/B + C)$ | How much is the sum of A divided by B and C? |
| 18 | $f = (A/B/C)$ | How much is A divided by B divided by C? |
| 19 | $f = (A/B - C)$ | How much is the difference between A divided by B and C? |
| 20 | $f = (A/B * C)$ | How much is A divided by B times C? |
| 21 | $f = (A - (B + C))$ | How much is A minus the sum of B and C? |
| 22 | $f = (A * B - C)$ | How much is the difference between A times B and C? |
| 23 | $f = (A/(B * C))$ | How much is A divided by the product of B and C? |
| 24 | $f = (A - B + C)$ | How much is A minus B plus C? |
| 25 | $f = (A + B + C)$ | How much is A plus B plus C? |
| 26 | $f = (A - B - C)$ | How much is A minus B minus C? |
| 27 | $f = (A * B/C)$ | How much is A times B divided by C? |
| 28 | $f = (A + B - C)$ | How much is A plus B minus C? |
| 29 | $f = (A * B * C)$ | How much is A times B times C? |

## H    Fine-tuning Details

For the one-operator data, we sample 500 examples for each of the 6 natural language templates in Table 2 and their corresponding original mathematical expressions, resulting in a total of 3,500 samples. For the two-operator data, we sample 1,000 examples for each of the 29 natural language templates in Table 3, resulting in a total of 29,000 samples. We set the maximum operand value to 100 and exclude problems with final results exceeding 1000. For single-operator data, we use an 8/2 train-test split, while for two-operator data, we use a 9/1 train-test split.

We fine-tune the OPT-1.3B model using the LoRA method [Hu et al., 2022] on one-operator and two-operator samples. For the one-operator templates, we train the model for 10 epochs with a batch size of 16. For the two-operator templates, we train the model for 20 epochs with a batch size of 32. The training uses a learning rate of 8e-4 with a linear decay scheduler. The LoRA configuration includes a rank of 8, a LoRA alpha of 32, and a dropout of 0.05.

## I    More experimental results

Figure 9 shows that LLMs with good arithmetic capabilities gradually focus more on high-order interaction patterns. Figure 10 illustrates that the OPT-1.3B model gradually increases its focus on

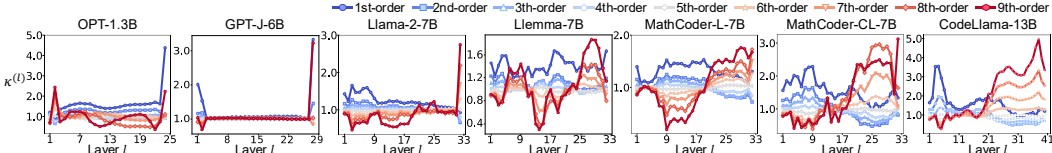

Figure 9: Comparing the normalized average strength $\kappa^{(l)}$ of interaction patterns of different orders encoded by LLMs during forward propagation. Each curve in the figure is averaged over various one-operator arithmetic queries, corresponding to template 4 in Table 2.

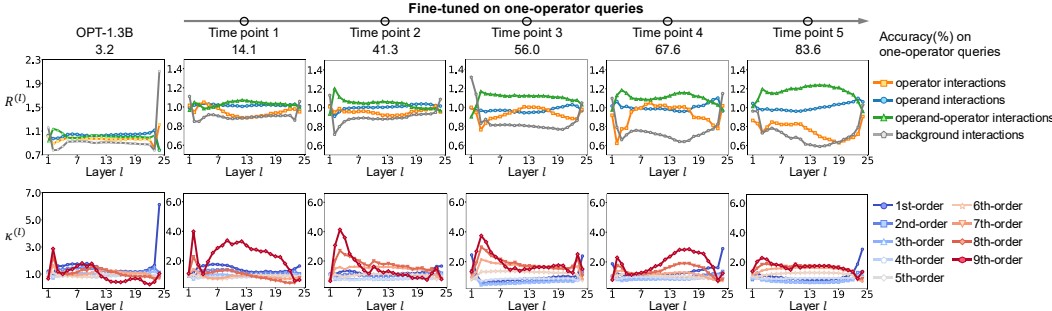

Figure 10: Visualizing the dynamic process of different interaction patterns encoded by the OPT-1.3B model during the training process. Each curve is averaged over various one-operator queries, corresponding to (a) template 0 and (b) template 3 in Table 2.

high-order interaction patterns during the learning process of simple arithmetic problems. Figure 11 demonstrates that the OPT-1.3B model gradually decreases its focus on high-order interactions while learning relatively complex arithmetic problems.

## J    Information about the use of AI assistants.

In this paper, AI tools such as DeepSeek were used for translation and grammar checking.

## K    Computational budget

We conducted our experiments on an NVIDIA GeForce RTX 4090 24GB GPU. For the Llama-2-7B model, the computation time per one-operator sample is around 30 seconds, while that for a two-operator sample is around 60 seconds.

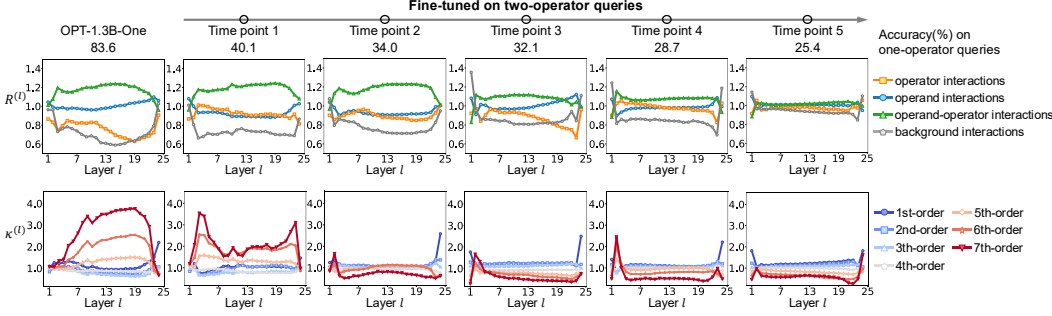

Figure 11: Visualizing the dynamic process of different interaction patterns encoded by the *OPT-1.3B-One* model during the training process. Each curve is averaged over various one-operator queries, corresponding to (a) template 0 and (b) template 3 in Table 2.

# L Limitations

When the number of input variables increases significantly, the computational cost of interaction analysis indeed rises. To deal with this issue, we can adopt the following approaches.

**Analyzing informative variables while treating uninformative variables as background** to reduce computational complexity while preserving core semantic information. Previous research [Chen et al., 2024] has demonstrated that selecting informative input variables does not fundamentally compromise the faithfulness of interactions.

**Phrase-level aggregation**, where related tokens are merged into meaningful phrases without compromising semantic integrity, can reduce the computational burden.

**Exploring approximation methods**, such as variable selection based on attention weights and prioritizing tokens that have a greater impact on model predictions for interaction analysis, can help reduce computational overhead.

# M Accuracy curve of the OPT-1.3B-One model on two-operator queries

We visualize the accuracy of the *OPT-1.3B-One* model on two-operator arithmetic queries across different checkpoints. As shown in Figure 12, the model demonstrates a gradual improvement in performance during the training process. We also highlight five selected checkpoints (referred to as TimePoints 1 through 5) that are used in Figure 7.

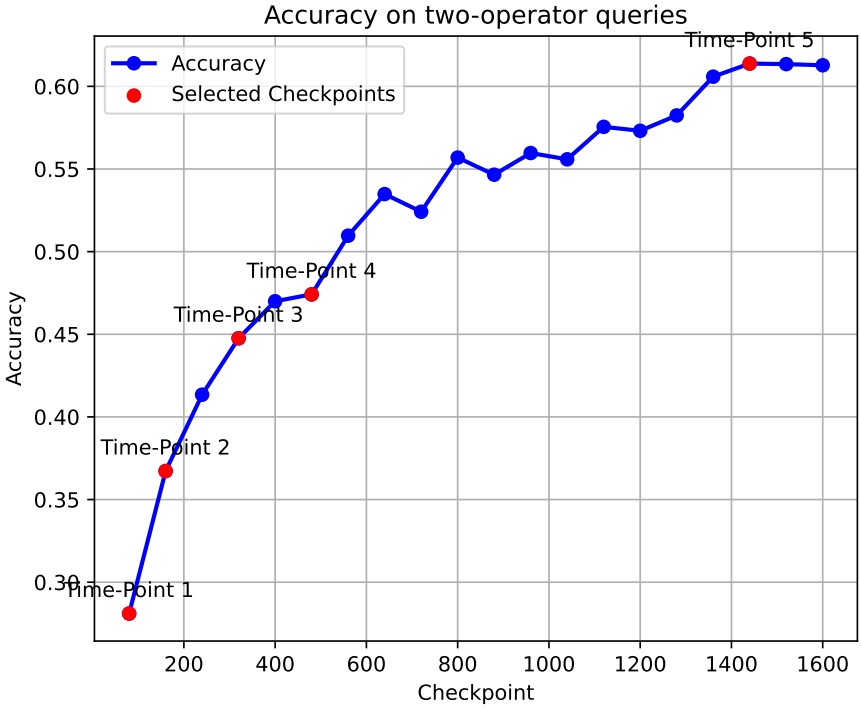

Figure 12: Accuracy of the *OPT-1.3B-One* model on two-operator queries evaluated at different checkpoints. Red markers indicate selected TimePoints 1–5 used in Figure 7.

