# OpenReview forum: "Interpreting Arithmetic Reasoning in Large Language Models using Game-Theoretic Interactions"
_NeurIPS.cc/2025/Conference — NeurIPS 2025 poster_

### Official Review · Reviewer_8cBX · 2025-06-24

**Clarity:** 4
**Significance:** 3
**Originality:** 3
**Rating:** 5
**Confidence:** 4

**Summary:**

Using game-theoretic measures, the paper explains arithmetic reasoning capabilities for LLMs through strengths of operand-operator interactions.

**Questions:**

- What is the motivation behind the cosine similarity as the choice for $v^{(l)}$, as opposed to say the mean squared difference?

- In Sec 4.1 the operand interactions do not seem to show  any consistent trends. What is the consequence of this behavior?

- Were embeddings considered at all? They could serve as a basic sanity check, since the tokens are not supposed to interact at the level of embeddings.

- It was not clear to me what the message of Appendix E was.

**Ethical Concerns:**

["NO or VERY MINOR ethics concerns only"]

**Final Justification:**

The paper answers my questions to adequate degree, and clarifies some subtle points not explicitly mentioned originally. However, it does not prompt me to increase my rating.

**Limitations:**

There were no particularly important limitations that stood out to me that was not addressed by the authors. Given that this is a preliminary investigation of game-theoretic measures on LLMs, there are lots of scope for future directions.

**Quality:**

3

**Strengths And Weaknesses:**

Strengths:

- The paper extends game-theoretic measures to LLMs, opening a different avenue for interpretability for such models.
- The experiments performed were quite extensive, and various evidence is provided to validate each of the claims made in the paper.
- The paper provides clear information regarding the template prompts, data sources, models chosen, etc.
- The metric chosen is easily generalizable to other settings.


Weaknesses:

- The paper studies the effect of each layer in one go, in other words, effects of attention heads are not studied separately. Given that the attention is primarily responsible for the token-token interaction, studying the cosine similarity of the head outputs could be illuminating.

- The score is averaged in a manner such that anti-parallel and parallel vectors are given equal importance. While I understand that without absolute values, the average would be "lower", it would be interesting to look at histogram or higher moments to understand if such weightage is appropriate.

---

> ### Author Rebuttal · Authors · 2025-07-30
>
> Thank you for your great efforts on the review of this paper. We will try our best to answer all your questions.
>
> $\color{blue}Q1:$ **"The paper did not study effects of attention heads separately. Given that the attention is primarily responsible for the token-token interaction, studying the cosine similarity of the head outputs could be illuminating."**
>
> **A:** Thank you. This is a valuable question. Following your advice, we **conducted new experiments to study effects of attention heads separately.** Specifically, we used **attention scores of the last token** to compute $v^{(l)}$ and then calculated the cosine similarity using this attention vector.
>
> We conducted experiments on all **32 (0-31) attention heads** of the MathCoder-L-7B model and observed consistent results across all of them: the model gradually focus more on **operand-operator interactions** (Table 1) as well as **higher-order interactions** (7th-order and 8th-order in Table 2), which aligns with our **Insight 1**. An example result for **head 15** is shown as follows.
>
> Table 1: Focality across **different types of interactions** at Layer $l$ using **head 15** outputs. (Mathcoder-L-7B)
>
> |Interaction Type|Layer 2|Layer 9|Layer 12|Layer 23|Layer 31|
> |-|-|-|-|-|-|
> |Operand|1.08|1.08|1.00|0.98|0.97|
> |Operator|0.99|0.72|0.94|0.94|0.89|
> |Operand-Operator|0.89|**1.16**|**1.07**|**1.08**|**1.14**|
> |Background|**1.10**|0.56|0.85|0.89|0.78|
>
> Table 2:  Focality across **interactions of different orders** at Layer $l$ using **head 15** outputs. (Mathcoder-L-7B template 1)
>
> |Interaction Order|Layer 2|Layer 9|Layer 12|Layer 23|Layer 31|
> |-|-|-|-|-|-|
> |1|**1.80**|0.48|0.96|0.84|0.89|
> |2|1.35|0.60|0.97|0.71|0.72|
> |3|1.16|0.79|0.97|0.74|0.73|
> |4|1.00|0.97|0.98|0.86|0.85|
> |5|0.83|1.17|1.02|1.08|1.11|
> |6|0.63|1.44|1.08|1.60|1.59|
> |7|0.49|1.81|1.18|2.31|2.38|
> |8|0.47|**2.30**|**1.31**|**3.19**|**3.06**|
>
> $\color{blue}Q2:$ **"The score is averaged in a manner such that anti-parallel and parallel vectors are given equal importance. It would be interesting to look at histogram or higher moments to understand if such weightage is appropriate."**
>
> **A:** Thank you. A interesting question! **To further investigate whether using the absolute value is appropriate, we analyzed the distribution of interaction effects.** Specifically, we computed the skewness (with zero as the axis of symmetry) of interaction effects at each layer. The results (Table 3) show that the maximum skewness is 0.033 (< 0.05), indicating that the distribution does not exhibit a strong bias toward either the positive or negative side. Therefore, treating positive and negative values with equal importance (i.e., using the absolute values) is reasonable.
>
> Table 3: Skewness (centered at 0) of interaction effects (MathCoder-CL-7B template 3).
>
> |Statistic|Layer 1|Layer 8|Layer 16|Layer 23|Layer 31|
> |-|-|-|-|-|-|
> |Skewness|0.008|0.006|**0.033**|0.014|-0.003|
>
> $\color{blue}Q3:$ **"What is the motivation behind the cosine similarity as the choice for $ v^{(l)}$ , as opposed to say the mean squared difference?"**
>
> **A:** Thank you for the thoughtful question. We chose **cosine similarity** for computing $v^{(l)}$ because it effectively captures the directional alignment between hidden representations. In Transformer-based architectures, semantic similarity is often encoded in the direction of vector representations rather than their magnitude. This makes cosine similarity **a natural and widely adopted choice**, not only in language modeling, but also in other domains such as recommender systems, where it has proven effective in measuring similarity between user and item embeddings.
>
> On the other hand, **mean squared difference (MSE)** measures absolute deviation in Euclidean space. While useful in some contexts, it conflates directional changes with scale differences, which may not be meaningful in the high-dimensional embedding spaces of LLMs.
>
> Nevertheless, we **conducted new experiments using the mean squared difference**. We replaced the cosine similarity with the mean squared difference, and normalized it to the range $[0,1]$:
>
> $v^{(l)}(x_T) = 1 - (MSE(f^{(l)}(x_T), f^{(l)}(x_N)) - d_{min}) / (d_{max} - d_{min} + ε)$
>
> **The experimental results confirm our main conclusion:** For simple one-operator arithmetic problems, LLMs that perform well (i.e., LLaMA-2-7B) tend to focus more on operand-operator interactions (Table 4) and high-order interactions (Table 5). While LLMs with weak performance (i.e., GPT-J-6B) tend to focus more on background interactions (Table 6) and extremely low-order interactions (Table 7).
>
> **While MSE and cosine similarity yield similar conclusions, we tend to adopt cosine similarity in this paper to avoid the impact of feature scale differences.**
>
> Table 4: Focality across **different types of interactions** at Layer $l$ using **MSE**. (LLaMA-2-7B)
>
> |Interaction Type|Layer 1|Layer 8|Layer 16|Layer 23|Layer 30|
> |-|-|-|-|-|-|
> |Operand|0.99|0.94|0.82|0.90|0.93|
> |Operator|0.94|0.98|0.90|0.71|0.67|
> |Operand-Operator|**1.10**|**1.10**|**1.31**|**1.34**|**1.33**|
> |Background|0.80|0.91|0.71|0.60|0.53|
>
> Table 5:  Focality  across **interactions of different orders** at Layer $l$ using **MSE**. (LLaMA-2-7B template 1)
>
> |Interaction Order|Layer 1|Layer 8|Layer 16|Layer 23|Layer 30|
> |-|-|-|-|-|-|
> |1|0.81|0.74|0.60|0.77|0.75|
> |2|0.49|0.38|0.26|0.34|0.39|
> |3|0.80|0.61|0.46|0.40|0.51|
> |4|1.02|0.93|0.79|0.60|0.72|
> |5|1.18|1.31|1.20|1.10|1.10|
> |6|1.38|1.70|2.01|2.25|2.02|
> |7|1.64|2.10|3.94|5.05|4.15|
> |8|**1.91**|**2.56**|**8.23**|**11.23**|**8.88**|
>
> Table 6: Focality across **different types of interactions** at Layer $l$ using **MSE**. (GPT-J-6B)
>
> |Interaction Type|Layer 1|Layer 6|Layer 11|Layer 17|Layer 23|
> |-|-|-|-|-|-|
> |Operand|0.79|0.71|0.71|0.74|0.77|
> |Operator|1.33|1.10|1.10|1.04|0.97|
> |Operand-Operator|0.81|0.69|0.72|0.77|0.82|
> |Background|**1.85**|**2.71**|**2.61**|**2.43**|**2.25**|
>
> Table 7:  Focality  across **interactions of different orders** at Layer $l$​ using **MSE**. (GPT-J-6B template 1)
>
> |Interaction Order|Layer 1|Layer 6|Layer 11|Layer 17|Layer 23|
> |-|-|-|-|-|-|
> |1|**4.94**|**2.71**|**2.27**|**2.00**|**1.95**|
> |2|1.18|0.82|0.81|0.82|0.81|
> |3|0.86|0.91|0.92|0.93|0.92|
> |4|0.77|0.96|0.98|0.99|0.99|
> |5|0.80|0.99|1.01|1.03|1.03|
> |6|0.93|1.02|1.04|1.04|1.06|
> |7|1.07|1.03|1.04|1.03|1.05|
> |8|1.92|0.97|0.98|0.92|0.95|
>
> $\color{blue}Q4:$ **"In Sec 4.1 the operand interactions do not seem to show any consistent trends. What is the consequence of this behavior?"**
>
> **A:** Thank you. That's a great question. Among the five LLMs that perform well on one-operator arithmetic problems (as shown in Table 8), the LLaMA-2-7B, Llemma-7B, and MathCoder-L-7B models show increasing focus on **operand interactions** in deep layers, while the CodeLlama-13B and MathCoder-CL-7B models do not show this trend. On the other hand, the CodeLlama-13B and MathCoder-CL-7B models show a more significant encoding of **operand-operator interactions** compared to the other three models. Among five LLMs, the LLaMA-2-7B model shows the weakest encoding of operand-operator interactions.
>
> The Llemma-7B and MathCoder-L-7B models achieve the highest accuracy on one-operator arithmetic tasks (Table 8), suggesting that their strong focus on operand interactions is beneficial for arithmetic reasoning. For the **LLaMA-2-7B** model, we believe that its limitation in encoding operand-operator interactions is mitigated by its stronger focus on operand interactions.
>
> As for the **CodeLlama-13B** and **MathCoder-CL-7B** model, although they do not tend to focus more on operand interactions, their strong modeling of **operand-operator interactions** is sufficient to support accurate arithmetic reasoning.
>
> Table 8: Accuracy (%) of LLMs on one-operator arithmetic tasks.
>
> |Model|1-opr Accuracy(%)|
> |-|-|
> |Llemma-7B|75.1|
> |MathCoder-L-7B|74.0|
> |CodeLlama-13B|71.1|
> |Llama-2-7B|65.1|
> |MathCoder-CL-7B|62.6|
>
> $\color{blue}Q5:$ **"Were embeddings considered at all? They could serve as a basic sanity check, since the tokens are not supposed to interact at the level of embeddings."**
>
> **A:** Thank you. This is an excellent question. As you correctly pointed out, tokens are not expected to interact at the embedding layer. Indeed, we verify this by computing interaction effects at the embedding layer, and **all values turn out to be zero**.
>
> $\color{blue}Q6:$ **"It was not clear to me what the message of Appendix E was."**
>
> **A:**  Thank you for pointing this out. Appendix E aims to justify our choice of using the last token's embedding to compute $v^{(l)}$, that is, the cosine similarity between embeddings of the original input and that of the masked input.
>
> To further illustrate this, we **conducted the same experiment** on the LLaMA-2-7B model with the prompt "*How much is 5 plus 1? Answer is .*" Results in Table 9 show that, except for the final token of the complete sentence, that is, token 10 ('*?*') and token 13 (the last token '*▁*'), the $v^{(l)}$ is nearly identical (close to 1). Only the embedding of the last token shows a significant change. Therefore, to simplify computation, we only focus on the change in the embedding of the last token, while ignoring the negligible changes in the embeddings of other tokens.
>
> Moreover, **several prior studies [Wendler et al. 2024] [Zhang et al. 2024] claim that in the Transformer architecture, the representation of last token includes information from all previous tokens.**
>
> Table 9: $v^{(l)}$ at different token positions across layers for the masked input "*How much is 5 [mask] [mask]? Answer is .*"
>
> |Token|Layer 5|Layer 20|Layer 30|
> |-|-|-|-|
> |token_1|0.00|0.00|0.00|
> |token_2|1.00|1.00|1.00|
> |token_3|1.00|1.00|0.99|
> |token_4|1.00|1.00|0.99|
> |token_5|1.00|1.00|0.99|
> |token_6|1.00|1.00|0.99|
> |token_7|1.00|1.00|1.00|
> |token_8|1.00|1.00|1.00|
> |token_9|1.00|1.00|1.00|
> |token_10|0.95|0.95|0.91|
> |token_11|1.00|1.00|0.99|
> |token_12|1.00|1.00|0.99|
> |token_13|**0.57**|**0.71**|**0.59**|

---

> > ### Comment · Reviewer_8cBX · 2025-08-05
> >
> > Thank you for your detailed response, which strengthens the claims made in an already strong paper. I will maintain my original score and recommend this paper to be accepted.

---

> > > ### Author Response · Authors · 2025-08-05
> > >
> > > We truly appreciate the time and effort you dedicated to reviewing our paper. Your valuable feedback has significantly enhanced the quality of our work.

---

### Official Review · Reviewer_5hoz · 2025-06-27

**Clarity:** 3
**Significance:** 3
**Originality:** 3
**Rating:** 4
**Confidence:** 4

**Summary:**

This paper introduces a game-theoretic framework to interpret the arithmetic reasoning capabilities of Large Language Models (LLMs). By disentangling the LLM's output score into contributions from various interactions between input words, the authors categorize these interactions into operand, operator, operand-operator, and background types, and also analyze their order (complexity). Through extensive experiments on one- and two-operator arithmetic problems across a range of LLMs, the study reveals that strong LLMs for simple problems primarily rely on operand-operator and high-order interactions, while for complex problems, they focus on operator and operand interactions. The paper also provides an interaction-based explanation for the task-specific nature of LoRA fine-tuning, demonstrating how it can lead to a decline in performance on simpler tasks by shifting the model's focus away from relevant interaction types.

**Questions:**

Please refer to the weakenesses

**Ethical Concerns:**

["NO or VERY MINOR ethics concerns only"]

**Final Justification:**

Thanks for the authors rebuttal, which addresses my most concerns. I keep my positive score.

**Limitations:**

yes

**Paper Formatting Concerns:**

N.A.

**Quality:**

3

**Strengths And Weaknesses:**

[Strengths]
The paper applies game-theoretic interactions (specifically Harsanyi interactions and derived focality metrics) to explain the internal mechanisms of LLMs for arithmetic reasoning. This provides an approach to interpret complex model behavior, moving beyond less rigorous neuron-level analysis or perturbation studies.
The classification of interactions into operand, operator, operand-operator, and background types, along with the analysis of interaction orders, offers a framework for understanding how LLMs process arithmetic queries.
The study visualizes how interaction patterns change during the fine-tuning process. This dynamic analysis reveals the learning trajectory of LLMs, showing how they adapt their focus on different interaction types and orders as they acquire new arithmetic skills, and also explains phenomena like LoRA's task-specific nature.

[Weaknesses]
The analysis of interactions in intermediate layers relies solely on the embedding of the last input token and its cosine similarity to the unmasked input. This is a simplification for complex transformer architectures, which distribute information across all tokens and layers via attention mechanisms. Might this choice lead to an incomplete or potentially biased understanding of how interactions are truly encoded and evolve within the model's internal states?

The paper acknowledges the absence of unified masking strategies and employs specific tokens (</s>, <|endoftext|>, <unk>) for masking. The choice of masking token may influence the model's internal representations and the resulting interaction values.

The study focuses exclusively on "simple arithmetic problems" (one- and two-operator queries) and explicitly states that it has not extended research to "more complex math word problems". This limits the generalizability of the findings.

The "focality" metric (Eq 8, 9) quantifies the strength of interaction types or orders by summing absolute interaction values. While useful for measuring overall "focus," this approach may disregards the direction (positive or negative contribution) of interactions, which is crucial for understanding the precise nature of the model's reasoning.

---

> ### Author Rebuttal · Authors · 2025-07-30
>
> Thank you for your great efforts on the review of this paper. We will try our best to answer all your questions.
>
> $\color{blue}Q1:$ **"The analysis of interactions in intermediate layers relies solely on the embedding of the last input token is a simplification for complex transformer architectures. Might this choice lead to an incomplete or potentially biased understanding?"**
>
> **A:** Thank you. **We have conducted experiments and found that the variation in middle-layer representations is primarily concentrated at the last token.** Please see Appendix E for detailed results and analysis. Therefore, to simplify computation, we only focus on the change in the embedding of the last token, while ignoring the negligible changes in the embeddings of other tokens.
>
> Nevertheless, **we show an example to support the above claim.** We tested a sample using the LLaMA-2-7B model: "*How much is 5 plus 1? Answer is*". This example contains 13 tokens: `['<s>', '▁How', '▁much', '▁is', '▁', '3', '▁times', '▁', '1', '?', '▁Answer', '▁is', '▁']`.
>
> We focused on a masked sentence: "*How much is 5 [mask] [mask]? Answer is ,*" Table 1 reports the cosine similarity between embeddings of the original input and that of the masked input ($v^{(l)}$). Results in Table 1 show that, except for the final token of the complete sentence, that is, token 10 ('*?*') and token 13 (the last token '*▁*'), the cosine similarity between the embeddings of the original input and the masked input is nearly identical (close to 1). Only the embedding of the last token shows a significant change.
>
> Moreover, **several prior studies [Wendler et al. 2024] [Zhang et al. 2024] claim that in the Transformer architecture, the representation of last token includes information from all previous tokens.**
>
> Table 1: $v^{(l)}$ at different token positions across layers for the masked input "*How much is 5 [mask] [mask]? Answer is ,*".
>
> |Token|Layer 5|Layer 20|Layer 30|
> |-|-|-|-|
> |token_1|0.00|0.00|0.00|
> |token_2|1.00|1.00|1.00|
> |token_3|1.00|1.00|0.99|
> |token_4|1.00|1.00|0.99|
> |token_5|1.00|1.00|0.99|
> |token_6|1.00|1.00|0.99|
> |token_7|1.00|1.00|1.00|
> |token_8|1.00|1.00|1.00|
> |token_9|1.00|1.00|1.00|
> |token_10|0.95|0.95|0.91|
> |token_11|1.00|1.00|0.99|
> |token_12|1.00|1.00|0.99|
> |token_13|**0.57**|**0.71**|**0.59**|
>
> $\color{blue}Q2:$ **"The paper acknowledges the absence of unified masking strategies and employs specific tokens may influence the model's internal representations and the resulting interaction values."**
>
> **A:** Thank you. A good question. We apply masking to remove the semantic meaning of the corresponding token, thereby effectively blocking its influence on the LLM's output.
>
> Since not all LLMs provide a *[pad_token]*, we use the model's *[unk_token]* as a placeholder, which represents an out-of-vocabulary token. At present, different LLMs offer different *[unk_token]* without specific semantic meaning, so we adopt different placeholders for different models accordingly.
>
> In recent years, some studies have proposed learning an optimal masking strategy for each model through training. However, such approaches are time-consuming, and thus we have not explored them in this paper. **Currently, there is no unified standard for masking, and we also highlight this point in Appendix A.**
>
> $\color{blue}Q3:$ **"The study focuses exclusively on "simple arithmetic problems", has not extended research to "more complex math word problems". This limits the generalizability of the findings."**
>
> **A:** Thank you. We **conducted new experiments** on a more complex math word problem: "*Henry has 1 notebook, then Henry receives 8 notebooks. What's the total number of notebooks that Henry has?*" This prompt is essentially an extension of a simple one-operator arithmetic problem "*1+8*=," but contains richer contextual information.
>
> We categorize input variables into 4 types: **numbers** (i,e., "*8*", "*1*"), **operators** (i.e., "*receives*", *"the total number of"*), entity words (including names and objects, i.e., "*Henry*", "*notebook*", "*notebooks*"), and background words (all words other than the above three categories).
>
> Following the same setting as in our paper, we also define four types of interactions:
>
> 1. **Operand interactions**: contain only numbers (no operators)
>
> 2. **Operator interactions**: contain only operators (no numbers)
>
> 3. **Operand-operator interactions**: contain both numbers and operators
>
> 4. **Background interactions**: contain neither numbers nor operators
>
> We  conducted experiments on the Mathcoder-L-7B model. **The results confirm our main conclusion (Insight 1)**: The internal mechanism of LLMs for solving simple one-operator arithmetic problems is their capability to encode operand-operator interactions (Table 2) and high-order interactions (Table 3) during the forward propagation.
>
> Table 2: Focality across **different types of interactions** at Layer $l$ on a free‐form math problem. (Mathcoder-L-7B)
>
> |Interaction Type|Layer 5|Layer 9|Layer 18|Layer 20|Layer 30|
> |-|-|-|-|-|-|
> |Operand|0.97|0.98|0.86|0.88|1.00|
> |Operator|1.10|**1.11**|1.09|1.04|0.97|
> |Operand-Operator|0.96|0.95|**1.13**|**1.16**|**1.02**|
> |Background|**1.11**|1.10|0.15|0.14|0.14|
>
> Table 3: Focality  across **interactions of different orders**  at Layer $l$ on a free‐form math problem. (Mathcoder-L-7B)
>
> |Interaction Order|Layer 5|Layer 9|Layer 18|Layer 20|Layer 30|
> |-|-|-|-|-|-|
> |1|**2.02**|**2.05**|2.10|1.86|1.31|
> |2|1.26|1.22|0.98|0.93|0.83|
> |3|1.06|1.03|0.85|0.77|0.79|
> |4|0.96|0.95|0.90|0.84|0.88|
> |5|0.88|0.89|0.99|1.03|1.07|
> |6|0.79|0.85|1.08|1.29|1.36|
> |7|0.67|0.83|1.52|1.96|1.91|
> |8|0.59|0.83|**3.10**|**3.85**|**3.13**|
>
> $\color{blue}Q4:$ **"The "focality" metric (Eq 8, 9) sums absolute interaction values. This approach may disregards the direction (positive or negative contribution) of interactions, which is crucial for understanding the precise nature of the model's reasoning."**
>
> **A:** Thank you. A good question! We **conducted new experiments** by dividing interactions into **positive** and **negative** categories.
>
> For different types of interactions:
> 1. Positive value: $R_{+}^{(l)}(\Omega^{\text{type}}) = \frac{\mathbb{E}_{S \in \Omega^{\text{type}},\ I_S^{(l)} > 0} I_S^{(l)}}{Z^{(l)}}$ (Table 4)
> 2. Negative value: $R_{-}^{(l)}(\Omega^{\text{type}}) = \frac{\mathbb{E}_{S \in \Omega^{\text{type}},\ I_S^{(l)} < 0} I_S^{(l)}}{Z^{(l)}}$ (Table 5)
>
> For interactions of different orders:
> 1. Positive value: $\kappa_{m,+}^{(l)} = \frac{\mathbb{E}_{|S|=m,\ I_S^{(l)} > 0} I_S^{(l)}}{Z^{(l)}}$ (Table 6)
> 2. Negative value: $\kappa_{m,-}^{(l)} = \frac{\mathbb{E}_{|S|=m,\ I_S^{(l)} < 0} I_S^{(l)}}{Z^{(l)}}$ (Table 7)
>
> **Results show that, in the later stage of forward propagation, the LLM shows the highest focus on operand-operator interactions (Table 4 and Table 5)  and high-order interactions (8th-order and 9th-order in Table 6 and Table 7). The results confirm our main conclusion (Insight 1)**: The internal mechanism of LLMs for solving simple one-operator arithmetic problems is their capability to encode operand-operator interactions and high-order interactions during the forward propagation.
>
> In summary, regardless of being positive or negative, interactions with large absolute values are considered to have a significant impact on the model’s output (i.e., whether pushing the prediction toward or away from the target answer, they still indicate strong influence).
>
> Table 4:   Focality across **different types of interactions** at Layer $l$ – **Positive values**.  (Mathcoder-CL-7b)
>
> |Interaction Type|Layer 1|Layer 8|Layer 16|Layer 23|Layer 31|
> |-|-|-|-|-|-|
> |Operand|1.00|1.03|1.01|0.88|0.84|
> |Operator|1.02|0.95|0.92|0.97|1.11|
> |Operand-Operator|1.00|0.96|**1.07**|**1.23**|**1.23**|
> |Background|**1.03**|**1.07**|0.94|0.77|0.87|
>
>
> Table 5: Focality across **different types of interactions** at Layer $l$ – **Negative values**.  (Mathcoder-CL-7b)
>
> |Interaction Type|Layer 1|Layer 8|Layer 16|Layer 23|Layer 31|
> |-|-|-|-|-|-|
> |Operand|-1.00|**-1.04**|-1.00|-0.88|-0.81|
> |Operator|**-1.02**|-1.00|-0.90|-0.96|-1.00|
> |Operand-Operator|-1.00|-0.98|**-1.06**|**-1.23**|**-1.23**|
> |Background|-0.99|-0.98|-0.88|-0.68|-0.74|
>
>
> Table 6: Focality across **interactions of different orders**  at Layer $l$ – **Positive values**.  (Mathcoder-CL-7b template 4).
>
> |Interaction Order|Layer 1|Layer 8|Layer 16|Layer 23|Layer 30|
> |-|-|-|-|-|-|
> |1|**1.56**|**2.27**|**1.43**|1.18|1.45|
> |2|0.00|0.46|0.00|0.64|0.30|
> |3|1.11|1.41|0.94|0.72|0.79|
> |4|0.00|0.61|0.00|0.87|0.38|
> |5|0.94|0.97|1.01|1.04|1.00|
> |6|0.00|0.70|0.24|1.07|0.65|
> |7|0.85|0.57|1.11|1.71|2.01|
> |8|0.00|0.74|0.33|1.39|1.97|
> |9|0.78|0.25|0.82|**2.40**|**2.34**|
>
> Table 7: Focality across **interactions of different orders**  at Layer $l$ – **Negative values**.  (Mathcoder-CL-7b template 4).
>
> |Interaction Order|Layer 1|Layer 8|Layer 16|Layer 23|Layer 30|
> |-|-|-|-|-|-|
> |1|0.00|0.00|0.00|0.00|0.00|
> |2|**-1.26**|**-1.66**|-1.00|-0.52|-0.79|
> |3|0.00|-0.54|0.00|-0.77|-0.27|
> |4|-1.01|-1.22|-0.95|-0.98|-0.83|
> |5|0.00|-0.70|-0.13|-1.00|-0.47|
> |6|-0.89|-0.77|**-1.07**|-1.21|-1.35|
> |7|0.00|-0.69|-0.17|-1.13|-0.88|
> |8|-0.81|-0.37|-1.03|**-2.37**|**-2.84**|
> |9|0.00|-0.81|-0.17|-1.89|-2.50|

---

> > ### Comment · Reviewer_5hoz · 2025-08-06
> >
> > Thanks for the authors rebuttal, which addresses my most concerns. I keep my positive score.

---

> > > ### Author Response · Authors · 2025-08-07
> > >
> > > We sincerely thank you for your time and effort in reviewing our work. We're glad that our responses have addressed your concerns, and we truly appreciate your positive evaluation.

---

### Official Review · Reviewer_XdQB · 2025-07-01

**Clarity:** 3
**Significance:** 3
**Originality:** 3
**Rating:** 4
**Confidence:** 3

**Summary:**

This paper introduces a game-theoretic method for studying inside transformer models when they do arithmetic. It shows that any LLM’s numeric output can be exactly written as the sum of Harsanyi interaction terms over all subsets of input tokens. By grouping them into “operand-only”, “operator-only”, “operand-operator”, and “background” categories, they find that strong models gradually amplify high-order, “operand-operator” interactions in late layers, thereby binding numbers to their operators. Using LoRA for fine-tuning, model gain on complex two-operator tasks at the cost of forgetting simpler one-operator reasoning, illustrating how tuning can reshape the collaboration patterns.

**Questions:**

1. Can you use other distance metrics (e.g. Euclidean) alongside cosine similarity to capture both angular and magnitude changes?
2. Can you provide a simple intervention study to produce the predicted jumps in interaction scores and output, thereby demonstrating causality? For example, flipping the operator token from ‘×’ to ‘+’.

**Ethical Concerns:**

["NO or VERY MINOR ethics concerns only"]

**Final Justification:**

The rebuttal has addressed my main concerns. First, the supplemented  Euclidean distance experiments confirmed the robustness of their findings beyond cosine similarity. Second, they proposed practical strategies with additional experiments to mitigate the scalability issue of Harsanyi interactions, which supports and verifies the feasibility. They also included the comparisons with Shapley value and an operator-flip intervention study, which strengthened the empirical support for their claims. Although the scalability remains a challenge, the additional results improve the paper's credibility. Therefore, I tend to borderline accept (rating 4) this paper.

**Limitations:**

yes

**Quality:**

3

**Strengths And Weaknesses:**

Strengths:
1. This paper is well-structured, with a solid theoretical foundation backed by rigorous game-theoretic proofs and clear mathematical derivations.
2. This paper provides layer-wise focality scores that clearly show how and when a model learns to bind numbers with operators. Besides, the methodology is model-agnostic and can be used on any transformer without changing its code.

Weaknesses:
1. The paper's empirical analysis is based on several assumptions; in my opinion, they are too strong to be fully convincing. The authors use cosine similarity (Eq.3) to quantify hidden-state changes across layers, which missed large magnitude shifts in a single dimension. Without metrics, like Euclidean distance, the analysis risks overlooking substantial activation shifts.
2. Harsanyi interactions require summing over 2^n subsets, which is infeasible when input length grows beyond a handful of tokens.
3. This paper does not compare its interaction decomposition with other interpretability or attribution techniques.

---

> ### Author Rebuttal · Authors · 2025-07-30
>
> Thank you for your great efforts on the review of this paper. We will try our best to answer all your questions.
>
> $\color{blue}Q1:$ **"The paper's empirical analysis is based on several assumptions which are too strong to be fully convincing. The authors use cosine similarity (Eq.3) to quantify hidden-state changes across layers, which missed large magnitude shifts in a single dimension. Without metrics, like Euclidean distance, the analysis risks overlooking substantial activation shifts."**
>
> **A:** Thank you. We would like to clarify whether the reviewer’s concern — *"they are several assumptions which are too strong to be fully convincing"* — refers specifically to our use of cosine similarity in Eq.3. If so, we **conducted new experiments** using **Euclidean distance**.
>
> We replaced the cosine similarity in the original Eq.3 with Euclidean distance, and normalized it to the range $[0,1]$:
>
> $v^{(l)}(x_T) = 1 - (EuclideanDistance(f^{(l)}(x_T), f^{(l)}(x_N)) - d_{min}) / (d_{max} - d_{min} + ε)$
>
> **The experimental results confirm our main conclusion:** For simple one-operator arithmetic problems, LLMs that perform well (i.e., the LLaMA-2-7B model) tend to focus more on operand-operator interactions (Table 1) and high-order interactions (Table 2) during forward propagation. While LLMs with weak performance (i.e., the GPT-J-6B model) tend to focus more on background interactions (Table 3) and extremely low-order interactions (Table 4).
>
> Table 1: Focality across **different types of interactions** at Layer $l$  using Euclidean distance (**LLaMA-2-7B**).
>
> |Interaction Type|Layer 1|Layer 8|Layer 16|Layer 23|Layer 31|
> |-|-|-|-|-|-|
> |Operand|1.01|0.91|0.78|0.84|0.96|
> |Operator|0.94|0.96|0.83|0.64|0.63|
> |Operand-Operator|**1.07**|**1.15**|**1.40**|**1.44**|**1.34**|
> |Background|0.82|0.88|0.63|0.52|0.48|
>
> Table 2:  Focality across **interactions of different orders** at Layer $l$  using Euclidean distance (**LLaMA-2-7B** template 1).
>
> |Interaction Order|Layer 1|Layer 8|Layer 16|Layer 23|Layer 31|
> |-|-|-|-|-|-|
> |1|0.76|0.72|0.51|0.62|0.82|
> |2|0.62|0.42|0.23|0.27|0.46|
> |3|0.83|0.62|0.42|0.35|0.60|
> |4|0.96|0.94|0.74|0.57|0.75|
> |5|1.12|1.28|1.16|1.09|1.04|
> |6|1.41|1.67|2.12|2.37|1.82|
> |7|1.75|2.03|4.38|5.41|3.81|
> |8|**2.14**|**3.56**|**10.88**|**13.67**|**10.14**|
>
> Table 3: Focality across **different types of interactions** at Layer $l$  using Euclidean distance (**GPT-J-6B**).
>
> |Interaction Type|Layer 3|Layer 9|Layer 15|Layer 21|Layer 27|
> |-|-|-|-|-|-|
> |Operand|0.62|0.66|0.69|0.72|0.68|
> |Operator|0.97|1.02|1.02|0.94|0.53|
> |Operand-Operator|0.63|0.67|0.72|0.77|0.74|
> |Background|**3.28**|**2.98**|**2.75**|**2.58**|**3.18**|
>
> Table 4:  Focality across **interactions of different orders** at Layer $l$ using Euclidean distance (**GPT-J-6B** template 1).
>
> |Interaction Order|Layer 3|Layer 9|Layer 15|Layer 21|Layer 27|
> |-|-|-|-|-|-|
> |1|**5.58**|**3.69**|**3.15**|**2.85**|**10.15**|
> |2|0.75|0.76|0.77|0.78|0.65|
> |3|0.82|0.87|0.89|0.90|0.72|
> |4|0.86|0.93|0.95|0.96|0.73|
> |5|0.89|0.97|0.99|1.00|0.72|
> |6|0.92|1.00|1.01|1.02|0.69|
> |7|0.95|1.03|1.00|1.01|0.61|
> |8|0.91|0.94|0.80|0.78|0.81|
>
> $\color{blue}Q2:$ **"Harsanyi interactions require summing over 2^n subsets, which is infeasible when input length grows beyond a handful of tokens."**
>
> **A:** Thank you. A good question! When the input length grows beyond a handful of tokens, the computational cost of interaction analysis indeed rises.
>
> To deal with this issue, we can adopt the following approaches.
>
> 1. **Analyzing informative variables while treating uninformative variables as background** to reduce computational cost while preserving core semantic information. Previous research [Chen et al., 2024] [Ren et al., 2024] [cite 1] has demonstrated that selecting informative input variables does not fundamentally compromise the faithfulness of interactions.
> 2. **Phrase-level aggregation**, where related tokens are merged into meaningful phrases without compromising semantic integrity, can reduce the computational cost.
>
> We have **conducted new experiments using the two approaches** described above. Specifically, we extended a simple one-operator arithmetic problem "*1+8*=" into a longer prompt:  "*Henry has 1 notebook, then Henry receives 8 notebooks. What's the total number of notebooks that Henry has?*" We categorize input variables into 3 types: **numbers** (i,e., "*8*", "*1*"), **operators** (i.e., "*receives*", *"the total number of"*), and **entity words** (including names and objects, i.e., "*Henry*", "*notebook*", "*notebooks*"). By using **Approach 1**, all other words not belonging to the above three categories are treated as background. By using **Approach 2**, the phrase *"the total number of"* is grouped as a single variable.
>
> Following the same setting as in our paper, we also define four types of interactions:
>
> 1. **Operand interactions**: contain only numbers.
>
> 2. **Operator interactions**: contain only operators.
>
> 3. **Operand-operator interactions**: contain both numbers and operators.
>
> 4. **Background interactions**: contain neither numbers nor operators.
>
> We conducted experiments on the Mathcoder-L-7B model. **The results confirm our Insight 1**: The internal mechanism of LLMs for solving simple one-operator arithmetic problems is their capability to encode operand-operator interactions  (Table 5) and high-order interactions (Table 6) during the forward propagation.
>
> Nevertheless, **we acknowledge that when $n$ is too large, the computation becomes infeasible. In fact, we have discussed this issue in the *Limitation* section of our paper.**
>
> Table 5: Focality across **different types of interactions** at Layer $l$ on a free‐form math problem. (Mathcoder-L-7B)
>
> |Interaction Type|Layer 5|Layer 9|Layer 18|Layer 20|Layer 30|
> |-|-|-|-|-|-|
> |Operand|0.97|0.98|0.86|0.88|1.00|
> |Operator|1.10|**1.11**|1.09|1.04|0.97|
> |Operand-Operator|0.96|0.95|**1.13**|**1.16**|**1.02**|
> |Background|**1.11**|1.10|0.15|0.14|0.14|
>
> Table 6: Focality across **interactions of different orders**  at Layer $l$ on a free‐form math problem. (Mathcoder-L-7B)
>
> |Interaction Order|Layer 5|Layer 9|Layer 18|Layer 20|Layer 30|
> |-|-|-|-|-|-|
> |1|**2.02**|**2.05**|2.10|1.86|1.31|
> |2|1.26|1.22|0.98|0.93|0.83|
> |3|1.06|1.03|0.85|0.77|0.79|
> |4|0.96|0.95|0.90|0.84|0.88|
> |5|0.88|0.89|0.99|1.03|1.07|
> |6|0.79|0.85|1.08|1.29|1.36|
> |7|0.67|0.83|1.52|1.96|1.91|
> |8|0.59|0.83|**3.10**|**3.85**|**3.13**|
>
> [cite 1] Li M. et al. Does a neural network really encode symbolic concepts? ICML, 2023.
>
> $\color{blue}Q3:$ **"This paper does not compare its interaction decomposition with other interpretability or attribution techniques."**
>
> **A:** Thank you. A good question! **Traditional interpretability and attribution methods**, such as Grad-CAM and Shapley values, often lack rigorous theoretical guarantees. In contrast, the interactions has been proven to be faithful explanations by a series of theoretical guarantees [Ren et al., 2023a]. The faithfulness of using interactions to explain an LLM is reflect as follows.
>
> 1. The output score of the LLM on any input sentence can be represented as the sum of the effects of all interactions.
> 2. The output score of the LLM on any masked sentence can be represented as the sum of the effects of all triggered interactions.
>
> Moreover, most attribution methods focus on the influence of individual input variable. However, a DNN does not independently use each variable for inference. Instead, a DNN relies on interactions among input variables for inference. For example, the interaction between "*green*" and "*hand*" forms the concept of "*beginner*," a meaning that cannot be derived from either word alone.
>
> Nevertheless, we **conducted new experiments using the Shapley value-based method** on the same single-operator templates. We computed Shapley values for different types of input variables during the forward propagation of the LLM. The results show that LLMs that perform well (i.e., the LLaMA-2-7B model) tend to focus most on operators, followed by operands, and lastly on background words (Table 7). However, **the Shapley value-based method fail to capture the joint effect of operands and operators on the model’s reasoning process.**
>
> Table 7: Average Shapley values for different types of input variable across layers. (LLaMA-2-7B)
>
> |Variable Type|Layer 1|Layer 8|Layer 16|Layer 23|Layer 31|
> |-|-|-|-|-|-|
> |Operand|**0.16**|0.09|0.14|0.17|0.16|
> |Operator|0.13|**0.15**|**0.17**|**0.18**|**0.18**|
> |Background|0.12|0.13|0.11|0.10|0.10|
>
>
> $\color{blue}Q4:$ **"Can you provide a simple intervention study to produce the predicted jumps in interaction scores and output, thereby demonstrating causality? For example, flipping the operator token from ‘×’ to ‘+’."**
>
> **A:** Yes, we **followed your advice to conduct experiments** to demonstrate the causality of the input operator variable (Table 8). Specifically, we input two prompts into the Mathcoder-L-7B model that differ **only in the operator** ("+" vs. "×"). **We observe that both the model’s output and the focality on different types of interactions change accordingly.**
>
> However, we cannot directly change the interaction effect to observe output changes, as it is an outcome of causal analysis based on how input changes lead to output differences, not a controllable variable in the process.
>
> Table 8: Changes in **model output** and **focality across different types of interactions at Layer $l$** when flipping the operator in arithmetic prompts. (Mathcoder-L-7B)
>
> **Prompt**: "*How much is 2 plus 4? Answer is* "
> **Model output**: "*6*"
>
> |Interaction Type|Layer 10|Layer 26|Layer 31|
> |-|-|-|-|
> |Operand|0.97|0.85|0.78|
> |Operator|1.02|0.66|0.76|
> |Operand-Operator|1.04|1.30|1.33|
> |Background|0.96|0.88|0.90|
>
> **Prompt**: "*How much is 2 times 4? Answer is* "
> **Model output**: "*8*"
>
> |Interaction Type|Layer 10|Layer 26|Layer 31|
> |-|--|--|--|
> |Operand|0.99|0.92|0.92|
> |Operator|0.98|0.89|0.99|
> |Operand-Operator|1.01|1.11|1.08|
> |Background|1.00|1.03|1.02|

---

> > ### Comment · Reviewer_XdQB · 2025-08-03
> >
> > Thanks for the detailed rebuttal and the additional experiments. Your responses addressed many of my earlier concerns. The work is indeed useful in advancing research on the interpretability of arithmetic reasoning in LLMs. I will increase my score accordingly.

---

> > > ### Author Response · Authors · 2025-08-03
> > >
> > > We sincerely thank you for your time and effort in reviewing our paper. Your helpful suggestions have greatly improved the clarity and quality of our work.

---

### Official Review · Reviewer_pSEa · 2025-07-01

**Clarity:** 3
**Significance:** 2
**Originality:** 3
**Rating:** 5
**Confidence:** 3

**Summary:**

This paper proposes explaining arithmetic reasoning in LLMs through game-theoretic interactions, quantifying operand-operator, operator-operand, and background interactions across layers. It identifies distinct interaction encoding patterns in LLMs for single-operator vs. multi-operator problems and links LoRA’s task-specificity to interaction shifts. This work provides a novel perspective for explaining arithmetic reasoning in LLMs, which may help users and developers understand, use, and improve LLMs.

**Questions:**

See the weaknesses.

**Ethical Concerns:**

["NO or VERY MINOR ethics concerns only"]

**Final Justification:**

In authors' rebuttal, my concerns have been well addressed.

**Limitations:**

Yes.

**Paper Formatting Concerns:**

N.A.

**Quality:**

3

**Strengths And Weaknesses:**

Strengths:
- Faithfulness guarantees via Harsanyi interactions provide a mathematically sound foundation, distinguishing it from heuristic attribution methods.
- The layer-wise analysis of interaction types effectively reveals distinct reasoning mechanisms for different problem types.
- Comprehensive experiments have been conducted to demonstrate the different types of interactions and validate the proposed insights.
Weaknesses
- Source code is not opened.
- The interaction effect can fit the output (Theorem 1), but since this is an interpretability study, whether the causal necessity can be verified?
- Experiments use templated arithmetic questions. It’s unclear how it performs in free‐form math problems.

---

> ### Author Rebuttal · Authors · 2025-07-30
>
> Thank you for your great efforts on the review of this paper. We will try our best to answer all your questions.
>
> $\color{blue}Q1:$ **"Source code is not opened."**
>
> **A:** Thank you. Due to the conference policy, we cannot provide any links during the rebuttal phase. The code will be released publicly upon acceptance.
>
> $\color{blue}Q2:$ **Ask for verify the causal necessity. "The interaction effect can fit the output, but since this is an interpretability study, whether the causal necessity can be verified?"**
>
> **A:** Thank you. This is a very good question. **Causal necessity** means that a certain factor is necessary for the model's prediction, i.e., without this factor, the model would not be able to produce the same output. **In our paper, the causal necessity is reflected in the influence of input variables on both the interaction effects and the LLM outputs.**
>
> In our paper, given an input $\boldsymbol{x}$ and an LLM, we can construct a **logical model** based on interactions to accurately fit the output of the LLM. The output of the logical model can be decomposed into the sum of the effects of all interactions in $\boldsymbol{x}$.
>
> Given an LLM, changing the input $\boldsymbol{x}$ leads to changes in the LLM's output. This process reflects the **causal necessity** of input variables, while interactions are the outcome derived from the above causal analysis. However, we cannot directly change interaction effects to influence the LLM's output.
>
> We **conducted new experiments** to demonstrate the causal necessity of the input variable (Table 1). Specifically, we input two prompts into the Mathcoder-L-7B model that differ **only in the operator** ("+" vs. "×"). **We observe that both the model’s output and the focality on different types of interactions change accordingly.**
>
> Table 1: Changes in **model output** and **focality across different types of interactions at Layer $l$** when flipping the operator in arithmetic prompts. (Mathcoder-L-7B)
>
> **Prompt**: "*How much is 2 **plus** 4? Answer is* "
> **Model output**: "*6*"
>
> |Interaction Type|Layer 10|Layer 26|Layer 31|
> |-|-|-|-|
> |Operand|0.97|0.85|0.78|
> |Operator|1.02|0.66|0.76|
> |Operand-Operator|1.04|1.30|1.33|
> |Background|0.96|0.88|0.90|
>
> **Prompt**: "*How much is 2 **times** 4? Answer is* "
> **Model output**: "*8*"
>
> |Interaction Type|Layer 10|Layer 26|Layer 31|
> |-|--|--|--|
> |Operand|0.99|0.92|0.92|
> |Operator|0.98|0.89|0.99|
> |Operand-Operator|1.01|1.11|1.08|
> |Background|1.00|1.03|1.02|
>
>
> $\color{blue}Q3:$ **Ask for new experiments on free-form math problems. "Experiments use templated arithmetic questions. It’s unclear how it performs in free‐form math problems."**
>
> **A:** Thank you. A good question. We have **conducted new experiments** on a free-form math problem : "*Henry has 1 notebook, then Henry receives 8 notebooks. What's the total number of notebooks that Henry has?*"
> We categorize input variables into 4 types: **numbers** (i,e., "*8*", "*1*"), **operators** (i.e., "*receives*", *"the total number of"*), **entity words** (including names and objects, i.e., "*Henry*", "*notebook*", "*notebooks*"), and background words (all words other than the above three categories).
>
> Following the same setting as in our paper, we also define four types of interactions:
>
> 1. **Operand interactions**: contain only numbers (no operators)
>
> 2. **Operator interactions**: contain only operators (no numbers)
>
> 3. **Operand-operator interactions**: contain both numbers and operators
>
> 4. **Background interactions**: contain neither numbers nor operators
>
> We  conducted experiments on the Mathcoder-L-7B model. We observe that during the forward propagation, the LLM’s focus on operand-operator interactions (see Table 2) gradually increases, along with its focus on higher-order interactions (7th-order and 8th-order in Table 3). **This is consistent with the Insight 1 in our paper. This experiment demonstrates that our method can be extended to free-form math problems.**
>
> Table 2: Focality across **different types of interactions** at Layer $l$ on a free‐form math problem. (Mathcoder-L-7B)
>
> |Interaction Type|Layer 5|Layer 9|Layer 18|Layer 20|Layer 30|
> |-|-|-|-|-|-|
> |Operand|0.97|0.98|0.86|0.88|1.00|
> |Operator|1.10|**1.11**|1.09|1.04|0.97|
> |Operand-Operator|0.96|0.95|**1.13**|**1.16**|**1.02**|
> |Background|**1.11**|1.10|0.15|0.14|0.14|
>
> Table 3: Focality  across **interactions of different orders**  at Layer $l$ on a free‐form math problem. (Mathcoder-L-7B)
>
> |Interaction Order|Layer 5|Layer 9|Layer 18|Layer 20|Layer 30|
> |-|-|-|-|-|-|
> |1|**2.02**|**2.05**|2.10|1.86|1.31|
> |2|1.26|1.22|0.98|0.93|0.83|
> |3|1.06|1.03|0.85|0.77|0.79|
> |4|0.96|0.95|0.90|0.84|0.88|
> |5|0.88|0.89|0.99|1.03|1.07|
> |6|0.79|0.85|1.08|1.29|1.36|
> |7|0.67|0.83|1.52|1.96|1.91|
> |8|0.59|0.83|**3.10**|**3.85**|**3.13**|

---

> > ### Comment · Reviewer_pSEa · 2025-08-06
> >
> > Thank the authors for their responses to my comments! My concerns have been well addressed. Therefore, I will adjust my score.

---

> > > ### Author Response · Authors · 2025-08-07
> > >
> > > We sincerely appreciate your thoughtful feedback, which has helped us improve the quality of our paper. We're glad to hear that your concerns have been well addressed.

---

### Decision · Program_Chairs · 2025-09-17

**Decision:**

Accept (poster)

**Comment:**

This paper presents a novel framework for interpreting arithmetic reasoning in LLMs through the lens of game-theoretic Harsanyi interactions. The authors provide strong theoretical grounding and extensive empirical analysis, including intervention studies, comparison with Shapley values, and experiments on both templated and free-form math problems. The work offers new insights into how LLMs encode operand-operator interactions and high-order interactions across layers, and explains task-specific behavior in LoRA fine-tuning. Therefore I recommend acceptance.